# Local PI(4,5)P$_2$ synthesis by septin-associated PIPKIγ isoforms controls centralspindlin association with the midbody during cytokinesis

Giulia Russo [1], Nadja Hümpfer [2], Nina Jaensch[2], Steffen Restel[1], Christopher Schmied [1], Florian Heyd [2], Martin Lehmann [1], Helge Ewers[2], Volker Haucke [1,2,3] ✉ & Michael Krauss [1] ✉

Cytokinesis critically depends on phosphatidylinositol 4,5-bisphosphate [PI(4,5)P$_2$]. Synthesis of PI(4,5)P$_2$ is crucial for several stages of cytokinesis, including actomyosin ring assembly and constriction, membrane tethering of spindle microtubules, and midbody organization. How these activities of PI(4,5)P$_2$ are spatiotemporally controlled is unknown. Here we unravel a crucial function for local PI(4,5)P$_2$ synthesis at the ingressed cleavage furrow by septin-binding isoforms of PIPKIγ to control midbody formation. We demonstrate that loss of PIPKIγ isoforms perturbs cytokinesis by impairing septin association with microtubules, and anillin and septin deposition at the intercellular bridge and at the midbody. This mechanism requires the ability of PIPKIγ isoforms to synthesize PI(4,5)P$_2$ and to associate with septins. Septins and PIPKIγ further synergize to promote centralspindlin recruitment to the midbody. Our findings establish septin-associated PIPKIγ isoforms as spatiotemporal controllers of midbody organization during cytokinesis that act through generating a local pool of PI4,5P$_2$ at the ingressed cleavage furrow.

Cytokinesis is the final step of cell division that ultimately partitions the cytoplasmic content of the mother cell into two daughter cells. Failures in cytokinesis can cause tetraploidy, centrosome amplification, and chromosomal instability, and thereby promote tumorigenesis[1]. Cytokinesis starts with the assembly of a contractile actomyosin ring at the equatorial plane of the mother cell that drives the formation of the cleavage furrow[2,3]. The cleavage furrow ingresses symmetrically, until the daughter cells are connected only by a thin intercellular bridge (ICB) formed around a tight bundle of central spindle microtubules[4], the cytokinetic bridge. As the ICB elongates, a dense structure assembles and matures in its center, the midbody, an organelle of complex and unique protein and lipid composition[5]. The midbody serves several important functions, including tethering of

spindle microtubules to the plasma membrane and the orchestration of the abscission machinery[6].

Cytokinesis is accompanied by drastic morphological changes of the mother cell, which rely on continuous remodeling and reorganization of the cytoskeleton. These events are orchestrated by phosphoinositides[7], in particular by the plasma membrane-enriched signaling lipid phosphatidylinositol 4,5-bisphosphate [PI(4,5)P$_2$][8]. PI(4,5)P$_2$ serves as a docking site for several components of the cytokinetic machinery[9]. It accumulates early at the newly forming cleavage furrow, where it anchors the contractile machinery to the cell equator to ensure symmetric ingression of the furrow[10]. This process is coordinated by anillin, a multidomain protein that scaffolds actin, myosin, septins, and the central spindle[11] and via its PI(4,5)P$_2$-binding pleckstrin

¹Leibniz-Forschungsinstitut für Molekulare Pharmakologie (FMP), Berlin, Germany. ²Faculty of Biology, Chemistry, Pharmacy, Freie Universität Berlin, Berlin, Germany. ³NeuroCure Cluster of Excellence, Charité, Universitätsmedizin, Berlin, Germany. ✉e-mail: haucke@fmp-berlin.de; krauss@fmp-berlin.de

homology (PH) domain attaches this scaffold to the plasma membrane[12,13]. Stable anchorage of anillin at the cleavage furrow also depends on RhoA[13], which is activated by microtubule-associated centralspindlin through Ect-2 at metaphase[14], and stabilized at the cortical membrane in an anillin- and PI(4,5)P$_2$-dependent fashion[15].

As the contractile ring closes, PI(4,5)P$_2$ further concentrates along the cleavage furrow, and later at the ICB to support bridge stabilization[10,16]. In addition to PI(4,5)P$_2$, ICB stability is also fostered by septins, a family of filament-forming, GTP-binding proteins[17,18] that are recruited to the cleavage furrow through direct interactions with anillin[10,12] and with PI(4,5)P$_2$. Septins are required for symmetric cleavage furrow ingression, and, by an unknown mechanism, promote stable anchorage of the ingressed furrow at the cytokinetic bridge to prevent furrow regression upon completion of contraction[19,20]. Septin filaments associate with membranes enriched in PI(4,5)P$_2$ in vitro and in living cells[21–23], and several lines of evidence indicate that PI(4,5)P$_2$ is required for the localization of septins at the ICB (reviewed in ref. 24).

At later stages of cytokinesis, when the midbody has formed, PI(4,5)P$_2$ tethers spindle microtubules to the plasma membrane through centralspindlin[25], a constitutive heterotetramer of MKLP1 (also known as KIF23) and MgcRacGAP (also known as RacGAP1)[26]. Whereas MKLP1 promotes bundling of antiparallel spindle microtubules at the midbody, MgcRacGAP locally controls the activities of Rho GTPases[27] and, through an atypical C1 domain, associates with plasma membrane PI(4,5)P$_2$ and phosphatidylinositol 4-phosphate [PI(4)P][25]. Coincidentally, PI(4,5)P$_2$ facilitates exocyst docking to the plasma membrane of the ICB, and thereby supports the abscission process[28].

An unsolved key conceptual problem is how the various distinct roles of the highly diffusible signaling lipid PI(4,5)P$_2$ during multiple stages of cytokinesis are controlled in space and time. In particular, it is unknown how PI(4,5)P$_2$ synthesis is mechanistically coupled to cytokinetic progression and midbody formation. One possibility is that distinct isoforms of type I PI(4)P 5-kinases, the main PI(4,5)P$_2$-synthesizing enzymes in mammalian cells, operate at distinct stages of cytokinesis and at defined sites. Type I PI(4)P 5-kinases are encoded by PIPKIα, Iβ, and Iγ genes, with several splice isoforms being known for PIPKIγ[29]. All isozymes are widely expressed in different tissues and cell types, but exhibit unique subcellular distributions, and trigger the formation of functionally distinct pools of PI(4,5)P$_2$. The mechanisms underlying the specific subcellular distribution of select isozymes remain incompletely understood. In the case of PIPKIγ, short splice inserts target kinase activity to distinct locations for defined subcellular events. For example, isoform i2 contains a short peptide that associates with AP-2 and talin to promote kinase recruitment to sites of endocytosis[30–32] and to focal adhesions[33–35], respectively. Isoform i5 localizes to endosomes[36] and autophagosomes[37], due to its interactions with SNX5 and Atg14.

PIPKIγ recruitment factors are often also PI(4,5)P$_2$ effectors, and, thus, considered part of a positive feedback loop that enhances PI(4,5)P$_2$ synthesis in their immediate proximity to reinforce their own membrane association[38]. It is likely that similar mechanisms underlie the recruitment of cytokinetic proteins, but up to now, no interactions with the PI(4,5)P$_2$-synthesizing machinery have been revealed.

Here, we close this fundamental knowledge gap regarding the mechanism of cell division by unraveling a function of PIPKIγ during ICB maturation and late midbody formation. We demonstrate that splice isoforms i3 and i5 associate with septins, and that this interaction is essential for the retention of anillin and septins at the ICB. We further show that the septin-dependent recruitment of PIPKIγ to the ICB is required for efficient targeting of centralspindlin to the midbody, and for the translocation of septins onto microtubules during late stages of cytokinesis. These findings establish septin-associated PIPKIγ isoforms as spatiotemporal controllers of midbody organization during cytokinesis that act through generating a local pool of PI4,5P$_2$ at the ingressed cleavage furrow. Given the signaling functions of postmitotic midbodies in regulating cell polarity and proliferation[39], these findings provide an avenue for future studies on how the septin/PIPKIγ-dependent assembly of the midbody affects its downstream signaling, also beyond cytokinesis.

## Results

### PIPKIγ exhibits a unique function at late stages of cytokinesis

The pivotal role of PI(4,5)P$_2$ during mitosis is well established, but little is known about the specific enzymes mediating its synthesis during cytokinesis. In mammalian cells, PI(4,5)P$_2$ is mainly synthesized by type I PIPK enzymes, named PIPKIα, Iβ, and Iγ. To identify the individual contributions of distinct type I PIPK enzymes, we depleted PIPKIα, Iβ, and Iγ isozymes in HeLa cells and subsequently analyzed the cells for cytokinetic defects. We immunostained cells for acetylated tubulin, a marker of spindle microtubules before furrow ingression, and of microtubules within the ICB from telophase until abscission (Fig. 1a). Approximately 2% of the cells displayed a mitotic spindle under control conditions (Fig. 1b; Supplementary Fig. 1a, b). Depletion of PIPKIβ, but not of PIPKIα, significantly increased this fraction, in line with previous reports[16]. Loss of PIPKIγ also raised the fraction of cells exhibiting an acetylated tubulin spindle, i.e., cells in meta- and anaphase. This was surprising as this isozyme is believed to contribute only a minor fraction of total PI(4,5)P$_2$ in non-neuronal cells[29]. More importantly, depletion of PIPKIγ, but not of Iα or Iβ, doubled the fraction of cells exhibiting an acetylated tubulin bridge, indicative of stalling in telophase (Fig. 1c). The resulting mitotic arrest led to an approximately threefold increase in the fraction of multinucleated cells (Supplementary Fig. 1c, d).

Our analyses also revealed other abnormalities in PIPKI-depleted cells, such as multipolar spindles or bridges. Loss of PIPKIβ or Iγ caused a significant increase in the fraction of cells that displayed a multipolar spindle (Fig. 1d; Supplementary Fig. 1e), while solely the depletion of PIPKIγ resulted in a significant increase in cells with multipolar bridges (Fig. 1e; Supplementary Fig. 1e). Depletion of neither type I PIPK isozyme significantly changed the overall PI(4,5)P$_2$ levels at the plasma membrane (Supplementary Fig. 1f, g). This was expected, as despite their unique tissue distribution and distinct subcellular localizations, PIPKIα, Iβ, and Iγ are able to largely compensate for each other[40] with respect to overall synthesis of PI(4,5)P$_2$.

These data reveal that while PIPKIβ and Iγ have synergistic roles during furrow ingression, PIPKIγ exhibits a hitherto unknown, unique function at late stages of cytokinesis, when the ICB is formed.

### PIPKIγ is required to concentrate the PI(4,5)P$_2$ effectors anillin and septins during ICB maturation

Given the unique defects observed upon depletion of PIPKIγ at late stages of cytokinesis, we analyzed the subcellular distribution of PI(4,5)P$_2$ effectors such as anillin in HeLa cells, that were synchronized by a thymidine/nocodazole block and fixed at a stage when they exhibit a mature ICB. Anillin bears a PI(4,5)P$_2$-binding PH domain in its C-terminus[11,12] that facilitates its recruitment to the equatorial plane of dividing cells. Upon closure of the contractile ring, anillin is found concentrated at the midbody, where it acts in concert with septins to anchor the midbody to the membrane[41], and—prior to abscission—follows septins to the midbody flanks[42]. Control cells or cells depleted of PIPKIα or Iβ displayed a compact organization of anillin at the ICB as visualized by co-staining for acetylated tubulin (Fig. 1f, g). In contrast, PIPKIγ knockdown cells lacked a compact organization of anillin (Fig. 1g), which instead was found scattered along the bridge. Additional clusters of anillin were detected in areas adjacent to the bridge, which were rarely observed under control conditions.

Through its C-terminal PH domain, anillin associates with septins[43], a family of guanine nucleotide-binding hetero-oligomerizing proteins, in a RhoA-dependent manner[44]. An additional septin binding

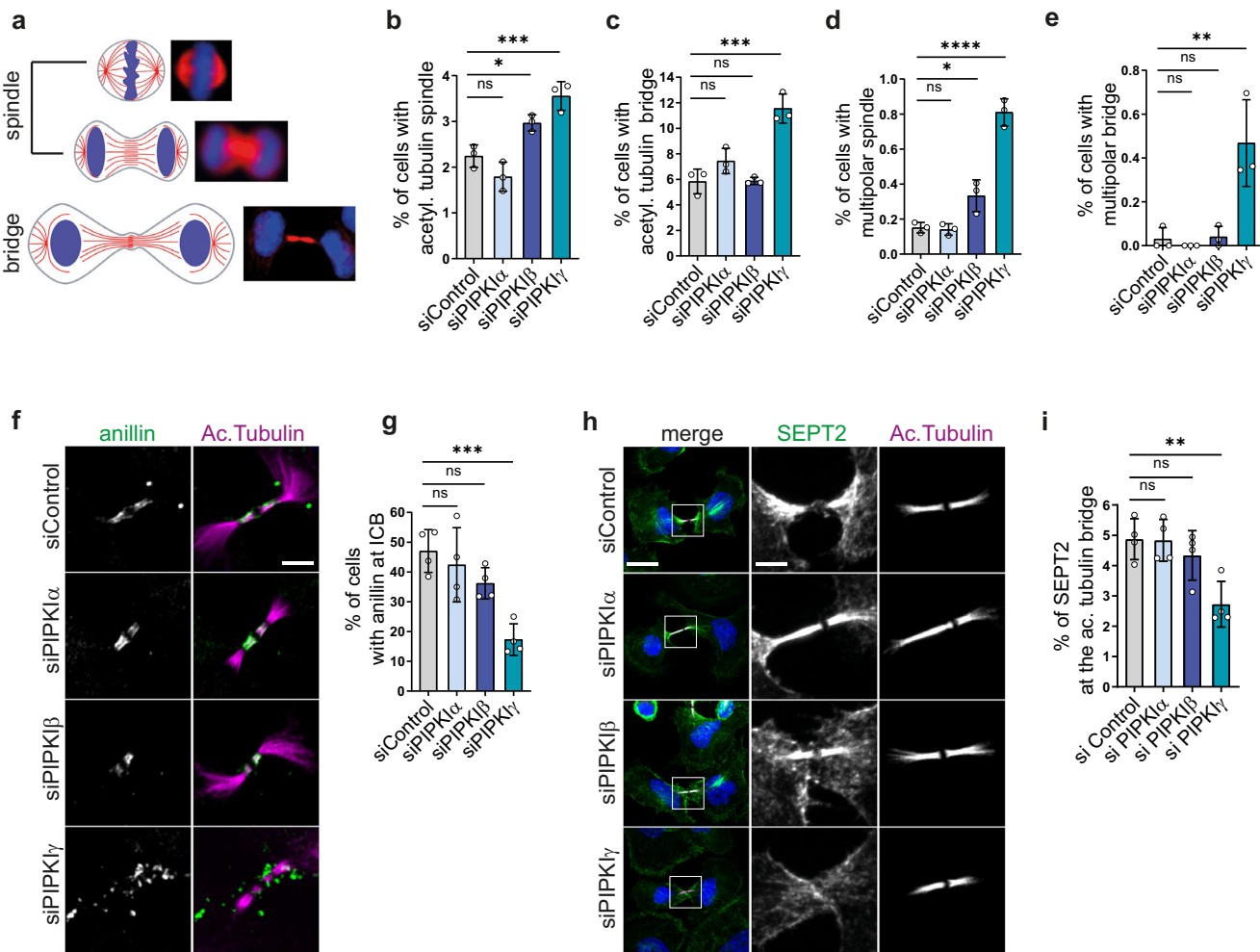

**Fig. 1 | PIPKIγ is required for cytokinetic progression and for the organization of anillin and septins at the ICB. a–e** HeLa cells were depleted of individual PIPKI isozymes, fixed and stained for acetylated tubulin, and analyzed by semi-automated imaging (see Supplementary Fig. S1b for exemplary images). Data are represented as mean ± SD ($n = 3$ independent experiments). Statistics: One-way ANOVA, followed by Dunnett's multiple comparison test. **a** Left: Scheme depicting the organization of acetylated microtubules (red lines) in mitotic cells before furrow ingression, and upon formation of the ICB. Right: Representative images of mitotic cells upon staining for acetylated tubulin (in red). **b** Percentage of cells displaying a central spindle, or **c** bundled microtubules at the ICB. **d** Percentage of cells with multipolar spindles, or **e** multipolar ICBs. **f, g** Depletion of PIPKIγ scatters anillin at the ICB. **f** Representative confocal images (max intensity z-projections) of HeLa cells treated with siRNA control, or against PIPKIα, β, γ, synchronized at late cytokinesis, and stained for acetylated tubulin and anillin. Note that the brightness of acetylated tubulin needed to be slightly increased to be able to visualize its localization upon knockdown of PIPKIγ. Scale bar: 5 µm. **g** Percentage of ICBs with anillin. Data are represented as mean ± SD ($n = 4$ independent experiments). Statistics: One-way ANOVA followed by Dunnett's multiple comparison test. **h, i** Depletion of PIPKIγ decreases the fraction of SEPT2 localizing at the acetylated tubulin bridge. **h** Representative confocal images (max intensity z-projection) of HeLa treated with siRNA control or against PIPKIα, β, γ, synchronized at late cytokinesis, and stained for acetylated tubulin and SEPT2. Scale bar of merge: 20 µm, scale bar of gray insets: 5 µm. **i** Percentage of total SEPT2 at the acetylated tubulin bridge. Values are represented as mean ± SD ($n = 4$ independent experiments). Statistics: One-way ANOVA, followed by Dunnett's multiple comparison test. *$P < 0.05$; **$P < 0.01$; ***$P < 0.001$; ****$P < 0.0001$. Source data and $P$-values are provided as a Source data file.

interaction surface is provided by the anillin N-terminus, which indirectly associates with SEPT9-enriched septins through CIN85[45]. PI(4,5)P₂ promotes septin association with membranes in living cells and in vitro[21–23]. Although largely dispensable during furrowing in mammalian cells, septins exert pivotal functions once the ICB is established and extends[42] by compartmentalizing select cytokinetic proteins at the midbody[19] and by scaffolding the assembly of the abscission machinery[42,46]. To analyze how individual PIPKI isozymes affect septin localization during cytokinesis, we assessed the subcellular distribution of SEPT2, a highly abundant septin isoform present in hexa- and octamers. In the vast majority of control cells, or in cells depleted of PIPKIα or PIPKIβ, SEPT2 was found to be enriched at the ICB (Fig. 1h) with about 5% of total cellular SEPT2 residing on the acetylated tubulin bridge (Fig. 1i). Depletion of PIPKIγ led to a marked loss of SEPT2 from ICB microtubules. Instead, SEPT2 remained associated with the plasma membrane of the newly forming daughter cells at sites adjacent to the bridge membrane (Fig. 1h, i).

These findings show that PIPKIγ spatiotemporally controls the organization of anillin and septins at the ICB.

## Distinct splice variants of PIPKIγ associate with septins and colocalize with septin filaments

The carboxyterminal region of human PIPKIγ undergoes alternative splicing, which gives rise to five isoforms termed PIPKIγ-i1-i5 (Fig. 2a). The distinct splice inserts promote PIPKIγ association with select binding partners at specific subcellular localizations[29] and this mechanism is thought to direct the generation of PI(4,5)P₂ at defined sites. To identify novel binding partners of these splice variants that might connect PIPKIγ to the cytokinetic machinery, we performed affinity purification experiments. Surprisingly, we discovered that

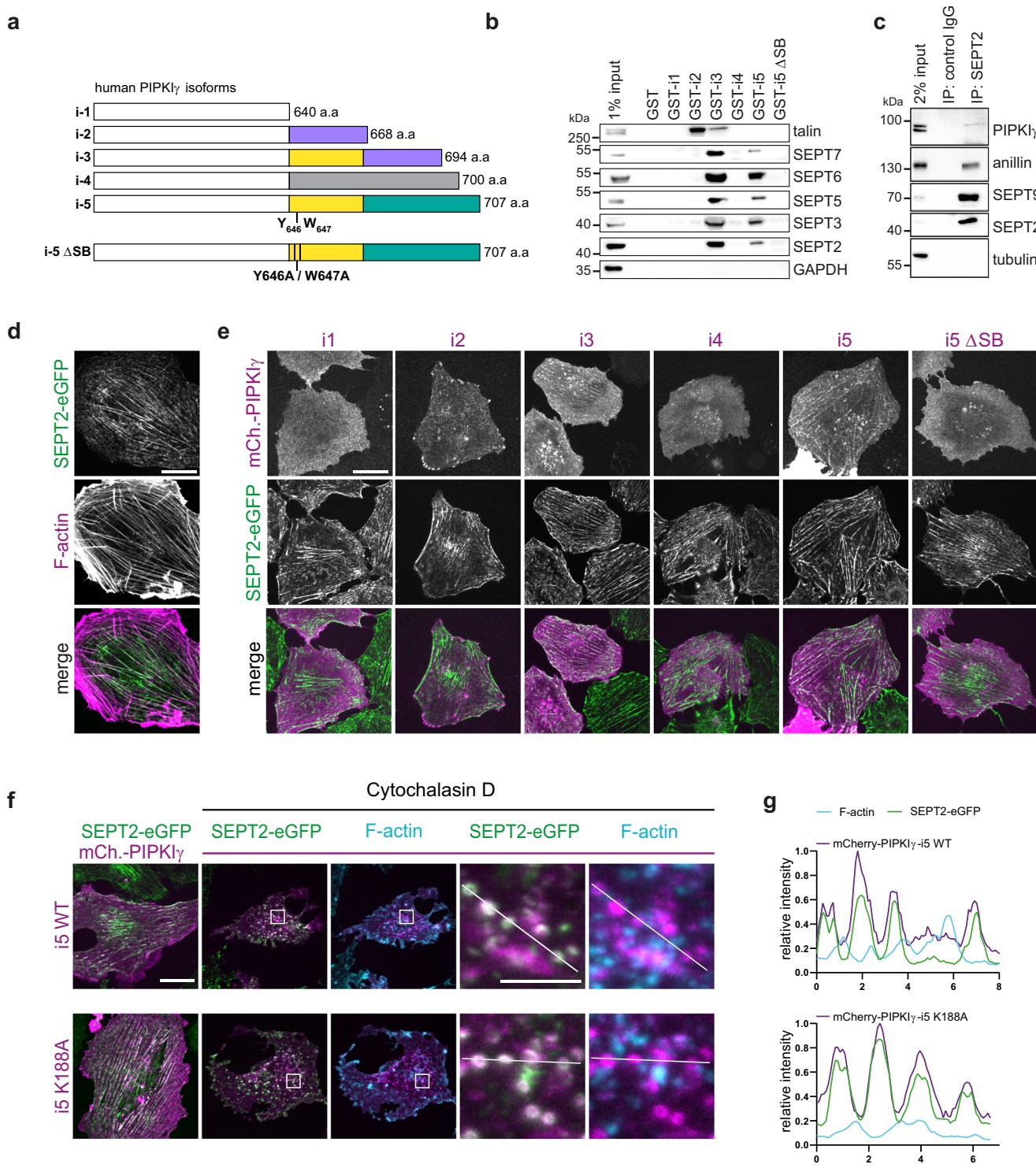

glutathione-S-transferase (GST)-fused carboxyterminal tails of both PIPKIγ-i3 and -i5 efficiently retained a variety of septin isoforms (Fig. 2b), indicative of a specific interaction of PIPKIγ-i3 and -i5 with septin oligomers. PIPKIγ-i2 and i3 efficiently associated with the focal adhesion protein talin from mouse brain lysates (Fig. 2b) as expected based on the presence of a talin binding-specific splice insert and on prior data[33–35] (Fig. 2a). Moreover, we found mCherry-PIPKIγ-i5 to co-immunoprecipitate myc-tagged SEPT7 or SEPT9 from HEK-293T cell lysates, but not endogenous talin (Supplementary Fig. 2a, b). PIPKIγ also co-purified with endogenous septins from extracts of mitotic HeLa cells (Fig. 2c). These data indicate that PIPKIγ isoforms undergo

complex formation with septins in living cells. We then performed alanine scanning mutagenesis along the i3/i5-specific splice insert and identified two neighboring residues (Y646 and W647) that are critically involved in septin binding: A point mutant variant of the PIPKIγ-i5 tail, i.e., Y646A/W647A, hereafter referred to as PIPKIγ-i5ΔSB ("deficient in septin binding"), failed to affinity-purify septins (Fig. 2b).

Next, we investigated a potential colocalization of PIPKIγ splice variants with septins. To this end, we transfected HeLa cells with HA-tagged versions of PIPKIγ. Notably, when expressed at low levels, HA-PIPKIγ-i3 and -i5 were organized in filaments that overlapped with endogenous septin fibers visualized by immunostaining of SEPT6

**Fig. 2 | PIPKIγ isoforms i3 and i5 specifically interact with septins. a–e** PIPKIγ isoforms 3 and 5 (i3/i5) interact with septins through two aromatic amino acids (W646 and Y647) harbored in their common splicing insert. **a** Schematic representation of C-terminal variations within the tail domains of human PIPKIγ isoforms. A splice insert present in isoforms i2 and i3 (purple) mediates talin binding. Isoforms i3 and i5 share a splice insert (yellow) of unknown function, and analyzed in detail in this study. Note that the kinase core domain common to all isoforms (white box) is not represented in scale. Unique splice inserts present in isoforms i4 and i5 are indicated in gray and green, respectively. **b** Affinity purification experiment on GST-fused kinase tails (aa451 to end) of human PIPKIγ isoforms. Material retained from mouse brain lysates was separated by SDS-PAGE and analyzed by Western blotting using the indicated antibodies. I5-ΔSB, double mutant (W646A/Y647A) defective in septin binding. **c** Septins affinity-purify together with PIPKIγ. Endogenous SEPT2 was immunoprecipitated from lysates of synchronized HeLa cells. The affinity-purified material was separated by SDS-PAGE and analyzed by Western blotting using the indicated antibodies. **d** Representative confocal image displaying partial overlap between endogenous SEPT2-eGFP and F-actin in NRK49F knock-in cells, stained with phalloidin. Scale bar: 20 μm. **e** Representative images of live NRK49F knock-in cells expressing SEPT2-eGFP upon transfection of mCherry (mCh.)-tagged PIPKIγ isoforms. Scale bar: 20 μm. **f, g** PIPKIγ-i5 association with septins is actin-independent, and does not rely on kinase activity. **f** Representative confocal images (max intensity z-projection) of NRK49F SEPT2-eGFP knock-in cells transfected with plasmids encoding human mCherry (mCh.)-tagged PIPKIγ-i5, or a kinase-dead mutant (K188A). Cells were incubated with 5 μM cytochalasin D to disrupt actin filaments. Cells were fixed, and F-actin was visualized by AF647-phalloidin. Scale bar: 20 μm, inset: 5 μm. **g** Intensity profiles (normalized to the maximum value) of F-actin, SEPT2-eGFP, and mCherry PIPKIγ-i5 wild type or K188A, along a line as depicted in (**f**). Source data are provided as a Source data file.

(Supplementary Fig. 2c). No such colocalization was observed for PIPKIγ-i1 or -i2, or for septin-binding-deficient mutant PIPKIγ-i5ΔSB. We corroborated these findings in genome-edited Norwegian Rat Kidney fibroblasts (NRK49F) that express SEPT2-eGFP under its endogenous promoter[47]. In this cell line, septins display a characteristic prevalent association with actin filaments (Fig. 2d). mCherry-PIPKIγ-i3 and -i5, but not septin-binding-deficient mutant PIPKIγ-i5ΔSB, colocalized with endogenous SEPT2 (Fig. 2e). PIPKIγ-i5 was also detected on intracellular organelles, in agreement with its known association with late endosomes and autophagosomes[36,37]. In contrast, mCherry-PIPKIγ-i1, i2, or i4, i.e., isoforms that do not bind to septins, were not colocalized with septin filaments. PIPKIγ-i1 was uniformly distributed across the plasma membrane, i2 additionally localized to focal adhesions[33,34], and i4 was detected in nuclear foci, presumably representing speckles[48].

F-actin is known to serve as a template for the assembly of septins into filaments[49]. Moreover, it has been shown that treatment with cytochalasin D releases septins from actin filaments and induces their reorganization into rings[49]. To rule out the possibility that the colocalization of septins with mCherry-PIPKIγ-i5 is bridged by actin, we treated NRK49F cells with cytochalasin D. Application of the drug triggered a rapid collapse of SEPT2-eGFP-containing filaments into rings, as seen before[20,49]. These rings retained their colocalization with mCherry-PIPKIγ-i5, but lost their association with F-actin (Fig. 2f, g; Supplementary Fig. 2d). Similar observations were made for catalytically inactive mutant mCherry-PIPKIγ-i5 (K188A).

Together, these findings demonstrate that PIPKIγ splice isoforms i3 and i5 associate with septins in living cells via a mechanism that is independent of lipid kinase activity or the association of septins with F-actin. Given the pivotal role of septins during cell division, these data suggest that the functions of PIPKIγ during late stages of cytokinesis may be exerted by its i3 and i5 splice variants.

### Spatiotemporally controlled PI(4,5)P₂ synthesis at the midbody by septin-associated PIPKIγ-i3/i5 promotes anillin recruitment

To directly test the role of PIPKIγ-i3/i5 in cytokinesis, we analyzed the subcellular distributions of PIPKIγ-i3/i5, anillin, and septins at different stages of mitosis. At anaphase, PIPKIγ-i5 was homogenously distributed at the plasma membrane of HeLa cells stably expressing mCherry-PIPKIγ-i5 (Fig. 3a). SEPT2 was enriched at the equatorial plane and at the poles of the mother cell, whereas anillin was found exclusively at the equatorial plane. During the transition to telophase, mCherry-PIPKIγ-i5 became concentrated at the cleavage furrow, where it aligned with SEPT2 and anillin (Fig. 3a). Upon completion of furrowing, mCherry-PIPKIγ-i5 localized to the ICB, where it outlined the midbody. At this stage, anillin accumulated at the midbody while SEPT2 started to translocate onto microtubules. At telophase, the distribution of mCherry-PIPKIγ-i3 at the midbody was indistinguishable from that of PIPKIγ-i5 (Supplementary Fig. 3a). This spatiotemporal choreography is consistent with a model in which PIPKIγ-i3/i5

executes a local function at the ICB (e.g., synthesis of PI(4,5)P₂) to promote the ingression of the cleavage furrow, possibly by enabling the recruitment of anillin and other factors.

To investigate whether the disorganization of anillin and septins at the ICB of PIPKIγ-depleted cells (Fig. 1f–i) reflects a selective loss-of-function of septin-binding isoforms of PIPKIγ, we depleted PIPKIγ-i3/i5 with an siRNA that targets the common splice insert[36]. This treatment reduced mRNA levels of the two cognate splice variants by about 85%, but did not affect the expression levels of other isoforms of PIPKIγ (Supplementary Fig. 3b, c). Select depletion of PIPKIγ-i3/i5 increased multinucleation to a similar degree as seen for the general depletion of all PIPKIγ transcripts (Supplementary Fig. 3d). This was accompanied by a dramatic scattering of anillin from the center of the ICB to its periphery (Fig. 3b, c). We also found that depletion of PIPKIγ-i5 alone caused a 1.9-fold increase in multinucleation, i.e., a phenotype milder than that elicited by combined loss of both PIPKIγ-i3 and -i5 (3.3-fold increase in multinucleation). This indicates that PIPKIγ-i3 is indeed expressed in HeLa cells, in agreement with an earlier report[50]. Moreover, these data suggest that PIPKIγ-i3 and PIPKIγ-i5 likely exhibit overlapping functions in cytokinesis in HeLa cells. Importantly, multinucleation as induced by depletion of PIPKIγ-i3/i5 was efficiently rescued in cells stably expressing siRNA-resistant, mCherry-tagged PIPKIγ-i5, but not in cells expressing mCherry (Supplementary Fig. 3e). Loss of PIPKIγ-i3/i5 did not affect the recruitment of anillin or septins to the cleavage furrow at earlier stages of cytokinesis (Supplementary Fig. 3f), emphasizing the predominant role of septin binding kinase isoforms during maturation of the ICB. Consistently, when monitoring microtubule dynamics in living cells upon application of SiR-tubulin we failed to observe significant changes in furrow ingression time upon depletion of PIPKIγ-i3/i5 (Supplementary Fig. 3g) Active myosin at the ingressed furrow, as evaluated by immunostainings of phospho-myosin light chain, was also unaltered (Supplementary Fig. 3i). By contrast, we observed a marked delay in the timing of microtubule severing (Supplementary Fig. 3h).

We hypothesized that anillin scattering from the ICB in cells depleted of PIPKIγ-i3/i5 might be a consequence of impaired local synthesis of PI(4,5)P₂. To test this, we first analyzed the impact of loss of PIPKIγ-i3/i5 on the levels and localization of PI(4,5)P₂ at the midbody of synchronized cells. In control cells PI(4,5)P₂ was found at the outlines of the ICB including regions adjacent to the midbody marked by citron kinase (Fig. 3d). Loss of PIPKIγ-i3/i5 resulted in a significant decrease in PI(4,5)P₂ at the midbody (Fig. 3d, e). Depletion of OCRL (oculocerebrorenal syndrome of Lowe protein), a PI(4,5)P₂ 5-phosphatase that is delivered to the ICB by Rab35 prior to abscission to locally erase PI(4,5)P₂[51], led to a more than two-fold increase in PI(4,5)P₂ levels along the ICB and at the midbody (Fig. 3d, e). The distribution of citron kinase remained unaffected by either knock-down condition (Fig. 3d, f). Next, we conducted genetic epistasis experiments to test if co-depletion of PIPKIγ-i3/i5 and the PI(4,5)P₂

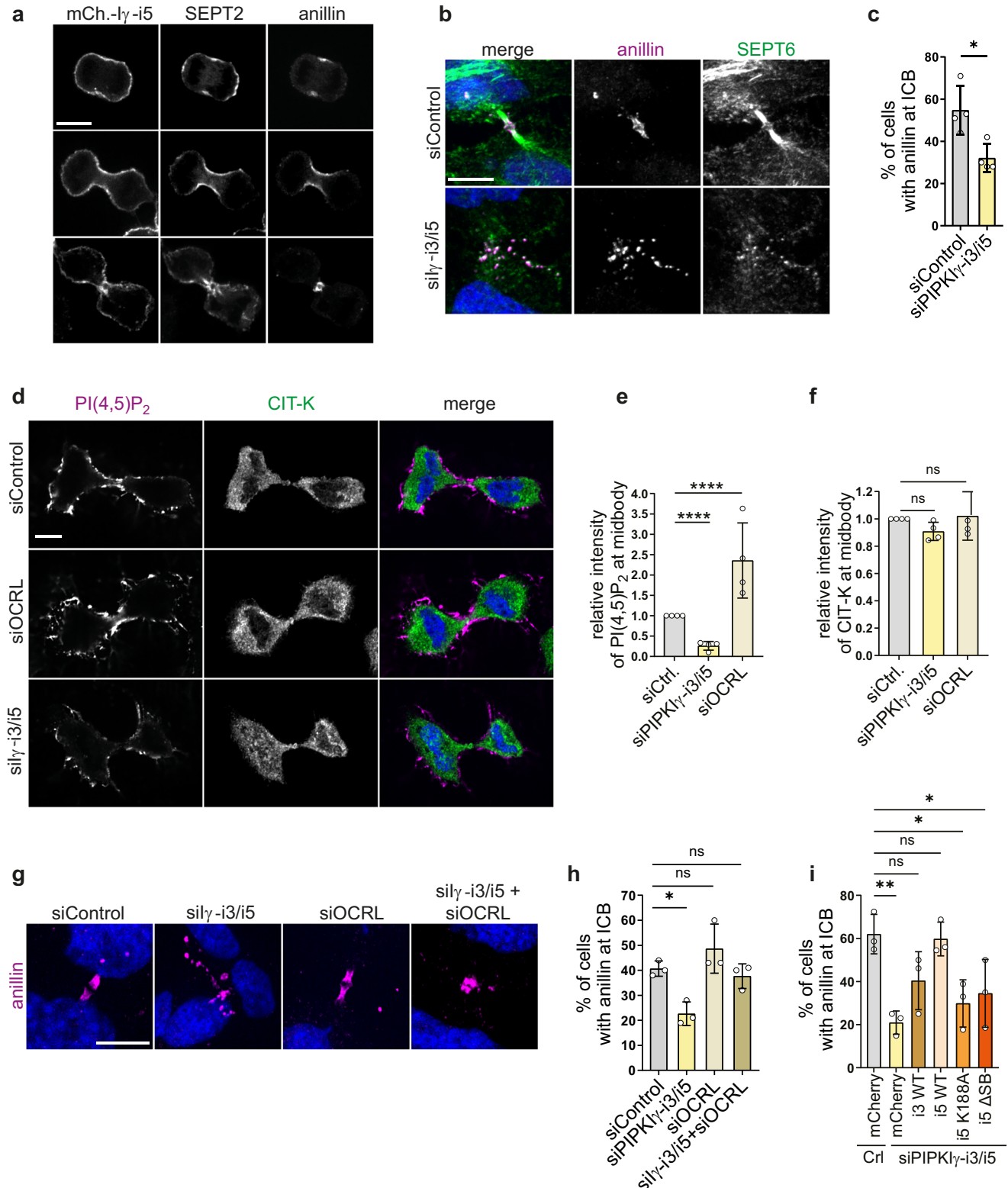

5-phosphatase OCRL can rescue defective anillin recruitment to the ICB. Knockdown of OCRL alone modestly increased the fraction of cells with anillin at their ICBs (Fig. 3g, h; Supplementary Fig. 3j). Co-depletion of OCRL together with PIPKIγ-i3/i5 rescued the mislocalization of anillin observed upon loss of PIPKIγ-i3/i5 alone. Third, we conducted rescue experiments using wild-type or catalytically inactive mutant PIPKIγ-i5, unable to synthesize PI(4,5)P$_2$. We observed that anillin concentration at the ICB was fully restored by reexpression of

mCherry-tagged, siRNA-resistant wild-type PIPKIγ-i3 or i5 (consistent with their overlapping functions, see above), but not by inactive mutant PIPKIγ-i5 (Fig. 3i, Supplementary Fig. 3k). Intriguingly, the septin-binding-defective mutant PIPKIγ-i5ΔSB also failed to rescue anillin dispersion. Failure to rescue anillin dispersion was associated with mistargeting of mutant PIPKIγ-i5: Whereas wildtype mCherry-PIPKIγ-i5 was found to be enriched at the ICB in close proximity to endogenous anillin (Fig. 3a, Supplementary Fig. 3k), mCherry-PIPKIγ-i5

**Fig. 3 | PIPKIγ-i3/i5 orchestrate local PI(4,5)P$_2$ synthesis to control anillin association with the midbody. a** Representative confocal images of HeLa cells stably expressing mCherry-PIPKIγ-i5. Cells were fixed at different mitotic stages, and stained for anillin and SEPT2, scale bar: 10 μm. **b** Representative confocal pictures (max intensity z-projection) of HeLa cells treated with siRNA against PIP-KIγ-i3/i5 or control, synchronized at late cytokinesis, and stained for anillin and SEPT6. Scale bar: 10 μm. **c** Percentage of ICBs with anillin. Data are represented as mean ± SD ($n = 4$ independent experiments). Statistics: two-tailed unpaired t-test. **d** Representative confocal images of HeLa cells upon depletion of PIPKIγ-i3/i5 or OCRL. Cells were synchronized at late stages of cytokinesis, and stained for PI(4,5)P$_2$ and CIT-K. Scale bar, 10 μm. **e** Relative intensity of PI(4,5)P$_2$ at the midbody. Normalized data are represented as mean ± SD ($n = 4$ independent experiments). Statistics: 1-way ANOVA, followed by Dunnett's multiple comparison test. **f** Relative intensity of CIT-K at the midbody. Normalized data are represented as mean ± SD ($n = 4$ independent experiments). Statistics: 1-way ANOVA, followed by Dunnett's

multiple comparison test. **g** Representative confocal images (maximal intensity z-projections) of anillin at ICBs of HeLa cells upon depletion of PIPKIγ-i3/i5, of OCRL, or of both. Cells at late stages of cytokinesis were immunostained for anillin. Scale bar, 10 μm. **h** Fraction of ICBs with anillin. Values indicate mean ± SD ($n = 3$ independent experiments). Statistics: 1-way ANOVA, followed by Dunnett's multiple comparison test. (see Supplementary Fig. 3j for knockdown validation). (**i**) Fraction of ICBs with anillin. PIPKIγ-i3/i5 was depleted in HeLa cells stably expressing mCherry or siRNA-resistant mCherry-tagged PIPKIγ-variants. The percentage of bridges displaying anillin was quantified and compared to HeLa cells stably expressing mCherry, and treated with control siRNA (see Supplementary Fig. 3k for representative images). Data are represented as mean ± SD ($n = 3$ independent experiments). Statistics: One-way ANOVA, followed by Dunnett's multiple comparison test. *$P < 0.05$; **$P < 0.01$; ****$P < 0.0001$. Source data and $P$-values are provided as a Source data file.

K188A, or ΔSB, were spread along the plasma membrane adjacent to the bridge, resembling the dispersed distribution of anillin (Supplementary Fig. 3k).

Taken together, these data suggest a model according to which septins anchor cognate PIPKIγ variants i3 and i5 at the midbody to produce a local pool of PI(4,5)P$_2$ that stably anchors anillin.

## PIPKIγ-i3/i5 is required for the organization of septins and microtubules at the ICB

Given the striking impact of septin-associated PIPKIγ-i3/i5 on midbody PI(4,5)P$_2$ levels and on anillin localization, we assessed their role in the dynamic reorganization of the septin cytoskeleton during cytokinesis. To this end, we generated a genome-edited cell line expressing eGFP-SEPT6 from its endogenous promoter. We obtained several heterozygous knock-in clones, in all of which endogenous SEPT6 and eGFP-SEPT6 protein levels decreased upon treatment with SEPT6-targeting siRNA (Supplementary Fig. 4a). Furthermore, eGFP-SEPT6 was successfully incorporated into septin filaments (Supplementary Fig. 4b) and displayed a high degree of colocalization with endogenous septin paralogs (Supplementary Fig. 4c), indicating that endogenous eGFP tagging does not interfere with SEPT6 function. Next, we monitored the subcellular distribution of eGFP-SEPT6 during cytokinesis (Supplementary Movie 1; representative frames reported in Supplementary Fig. 4d). Under control conditions, eGFP-SEPT6 first accumulated at the cleavage furrow (white asterisk in Supplementary Fig. 4d). About 30 min after completion of furrow ingression the ICB appeared. At the same time, eGFP-SEPT6 started to relocate from the cell cortex onto putative microtubules within the ICB and remained there during bridge elongation until abscission. After abscission the daughter cells displayed perinuclear, sinuous septin fibers that likely derived from the ICB (white arrowheads). In cells depleted of PIPKIγ-i3/i5 eGFP-SEPT6 retained its initial localization at the cleavage furrow but failed to accumulate at the ICB and to translocate onto microtubules. Moreover, the resulting daughter cells lacked the prominent septin fibers observed under control conditions. This defect persisted throughout interphase (Supplementary Movie 2; representative frames reported in Supplementary Fig. 4d). In addition, we observed that about 29.2% (±8.4% SEM, $n = 3$) of kinase-depleted cells died during cell division, compared to 6.6% (±4.3% SEM, $n = 3$) of control cells. As a result, only about 60.6% (±3.5% SEM, $n = 3$) of knockdown cells underwent successful division, compared to 82.7% (±6.5% SEM, $n = 3$) of control cells.

Septins are well known to associate with microtubules to regulate their stability and dynamics[18]. Given the dramatic impact of PIPKIγ on septin distribution, we investigated the organization of ICB microtubules in detail. Depletion of PIPKIγ-i3/i5-induced scattering of septin filaments away from the acetylated tubulin bridge (Fig. 4a, b), similar to cells lacking all PIPKIγ isoforms (compare Fig. 1h). Furthermore, microtubule stability assessed by acetylated tubulin was impaired

(Fig. 4a, c). Loss of PIPKIγ-i3/i5 also significantly reduced the length of the acetylated tubulin bridge (Fig. 4d) and microtubule bundling along the ICB (Fig. 4a,e). Consistent with the latter observation, ICBs of kinase-depleted cells accumulated significantly lower levels of the microtubule bundling factor PRC1 (Supplementary Fig. 4e, f). The defective septin organization at the ICB (Fig. 4f, Supplementary Fig. 4g), the reduced length of the acetylated tubulin bridge (Fig. 4g, Supplementary Fig. 4g), and the defective microtubule bundling (Fig. 4h, Supplementary Fig. 4g) were rescued by expression of wild-type mCherry-PIPKIγ-i3 or -i5, but not of -i5 mutants lacking catalytic activity or the ability to associate with septins.

In summary, these data demonstrate that PIPKIγ-i3/i5 isoforms are required for the organization of septins and microtubules at the ICB via a mechanism that depends on their ability to locally synthesize PI(4,5)P$_2$ and to bind to septins.

## Septin-dependent recruitment of PIPKIγ is essential to maintain centralspindlin at the midbody

The observed impact of PIPKIγ-i3/i5 depletion on microtubule organization and stability prompted us to analyze the effect of PIPKIγ-i3/i5 loss on midbody-associated microtubule-binding proteins, in particular on centralspindlin. The centralspindlin protein complex is composed of MgcRacGAP and mitotic kinesin-like protein 1 (MKLP1) and tethers midbody microtubules to the plasma membrane[26]. MgcRacGAP comprises an atypical C1-domain with affinity for PI(4,5)P$_2$[25]. Given the impact of PIPKIγ-i3/i5 depletion on septin and microtubule organization during cytokinesis, we hypothesized that the septin-dependent recruitment of PIPKIγ is required for the stable association of centralspindlin with the cell cortex. To test this hypothesis, we assessed the localization of centralspindlin at late stages of cytokinesis. As expected, under control conditions, MKLP1 colocalized with the midbody ring visualized by immunostaining for citron kinase (Fig. 5a). Notably, knockdown of PIPKIγ-i3/i5 had no impact on the distribution of citron kinase or on its abundance at the midbody (Fig. 5a, b). Loss of PIPKIγ-i3/i5, thus, does not generally affect midbody composition or positioning. In contrast, depletion of PIPKIγ-i3/i5 significantly decreased MKLP1 levels at the midbody ring (Fig. 5a, c). This defect was rescued by expression of wild-type mCherry-PIPKIγ-i5, but not catalytically inactive or septin-binding-deficient kinase mutants (Fig. 5d, Supplementary Fig. 5a). Depletion of SEPT2 impaired the deposition of centralspindlin at the midbody to an extent similar to that observed upon loss of PIPKIγ-i3/i5 (Fig. 5e, f). These results are consistent with a primary role of septins in guiding PIPKIγ to the ICB.

Next, we aimed at gaining insight into the spatial organization of septins, microtubules, and centralspindlin at the midbody. The detection of midbody proteins by conventional fluorescence microscopy is complicated by the dense packing of individual components in this organelle, which allows only limited access for antibodies. To solve this problem, we used ultrastructure expansion microscopy (U-

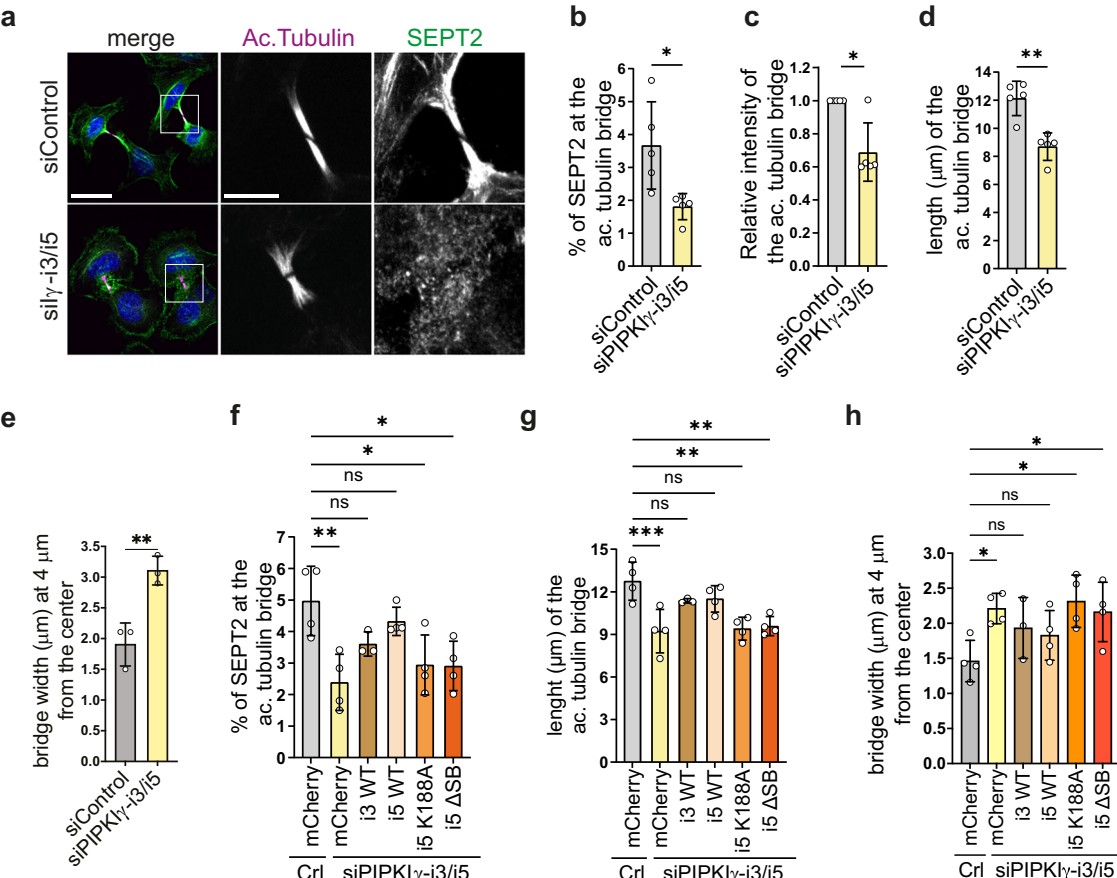

**Fig. 4 | Loss of PIPKIγ-i3/i5 perturbs ICB integrity and impairs SEPT2 recruitment to the ICB. a** Representative confocal images (max intensity z-projection) of HeLa cells treated with siRNA against PIPKIγ-i3/i5 or control, synchronized at late cytokinesis, and stained for acetylated tubulin and SEPT2. Scale bar of merge: 30 μm, of gray insets: 10 μm. **b** Percentage of total SEPT2 detected at the acetylated tubulin bridge, depicted as mean ± SD ($n = 5$ independent experiments). Statistic: two-tailed, unpaired t-test. **c** Relative intensity of the acetylated tubulin bridge. Normalized data are represented as mean ± SD ($n = 5$ independent experiments). Statistics: two-tailed, one sample t test. **d** Length of the acetylated tubulin bridge, measured as max Feret diameter of the ROI delimiting the acetylated tubulin bridge. Data are represented as mean ± SD ($n = 5$ independent experiments). Statistics: two-tailed, unpaired t-test. **e** Width of the acetylated tubulin bridge at a distance of 4 μm from the center. Data are represented as mean ± SD ($n = 3$ independent experiments). Statistics: two-tailed, unpaired t-test. **f** Percentage of total

SEPT2 at the acetylated tubulin bridge in knockdown cells stably expressing the indicated PIPKIγ-variants (see Supplementary Fig. 4g for representative images). Bar diagram depicts mean ± SD ($n \geq 3$ independent experiments). Statistics: One-way ANOVA, followed by Dunnett's multiple comparison test. **g** Length of the acetylated tubulin bridge in knockdown cells stably expressing the indicated PIPKIγ-variants. Data depict mean ± SD ($n \geq 3$ independent experiments; see Supplementary Fig. 4g for representative images.) Statistics: One-way ANOVA, followed by Dunnett's multiple comparison test. **h** Width of the acetylated tubulin bridge at a distance of 4 μm from the center in knockdown cells stably expressing the indicated PIPKIγ-variants. Data depict mean ± SD ($n \geq 3$ independent experiments; see Supplementary Fig. 4g for representative images). Statistics: One-way ANOVA, followed by Dunnett's multiple comparison test. *$P < 0.05$; **$P < 0.01$; ***$P < 0.001$. Source data and $P$-values are provided as a Source data file.

ExM)[52], for which the specimen is physically expanded prior to antibody labeling[53]. At cytokinesis control cells exhibited bundles of antiparallel microtubules along the ICB that overlapped at the midbody (Supplementary Fig. 5b). Septins were not found at the midbody itself, but formed two rings that aligned with two secondary ingression sites at its flanks (Supplementary Fig. 5b). Additionally, septins colocalized with microtubules that protruded away from the secondary ingression sites into the developing daughter cells. Upon depletion of PIPKIγ-i3/i5, septins underwent a dramatic redistribution: they were mostly dispersed from the ICB and only sparsely aligned with microtubules (Supplementary Fig. 5b). Instead of being organized in continuous straight fibers, they formed short rods and curly filaments in proximity to the ICB at the periphery of the daughter cells. We then assessed the distribution of centralspindlin by U-ExM. In control cells, centralspindlin localized to a confined region at the midbody where microtubules overlapped. In PIPKIγ-i3/i5-depleted cells, total centralspindlin levels at this locale were significantly reduced, akin to our findings in non-expanded specimens (Fig. 5g).

The PI(4,5)P$_2$-dependent recruitment of effector proteins to the plasma membrane is often fostered by complex formation between an effector protein and a PI(4,5)P$_2$-synthesizing enzyme to locally boost PI(4,5)P$_2$ synthesis[38]. This raises the possibility that centralspindlin might interact with PIPKIγ. To test this, we screened for binding partners of septins and PIPKIγ in cytokinetic HeLa cells. We identified the centralspindlin component MKLP1 in immunoprecipitates of endogenous SEPT2 and of endogenous PIPKIγ (Fig. 5h, i). Citron kinase, a known interactor of centralspindlin[54], was detected in both immunoprecipitates as well. Anillin efficiently copurified with septins, but not with PIPKIγ. Hence, the association of septins with PIPKIγ is not bridged by anillin. Furthermore, the microtubule bundling factor PRC1 reliably precipitated with SEPT2 and recruited septins onto hyperbundled microtubules in transfected fibroblasts (Supplementary Fig. 5c), suggesting a role of the microtubule bundling machinery in septin recruitment onto microtubules.

Next, we investigated how protein levels of PIPKIγ might be affected by key partners identified in this study. Depletion of anillin or

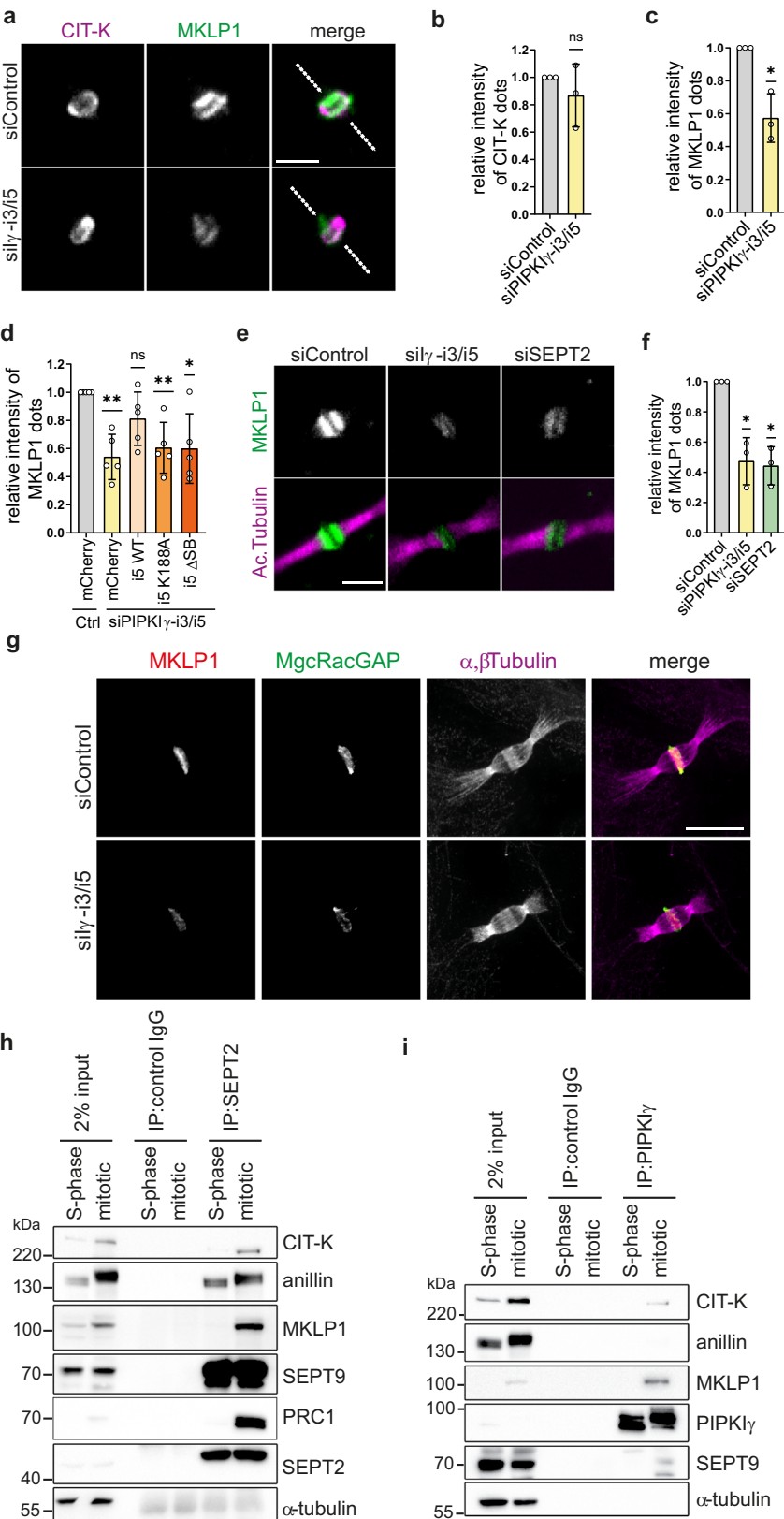

of MgcRacGAP led to significantly decreased levels of PIPKIγ (Supplementary Fig. 5d), whereas knockdown of SEPT7, a subunit required for maintaining septin cytoskeleton integrity, did not affect PIPKIγ levels. This indicates that anillin-[41] and MgcRacGAP-[25]mediated midbody tethering to the plasma membrane is important for PIPKIγ stability. Finally, we depleted anillin, McgRacGAP, or SEPT7 in stably

transfected, synchronized HeLa cells to monitor the subcellular distribution of mCherry-tagged PIPKIγ-i5 (WT). In control cells, mCherry-PIPKIγ-i5 prominently outlined late-stage ICBs (Supplementary Fig. 5e), as expected. Cells depleted of anillin, MgcRacGAP, or SEPT7 showed severe mitotic defects, consistent with previous reports[19,25,43], and only a few cytokinetic cells displayed an elongated acetylated tubulin

**Fig. 5 | A septin-interacting PIPKIγ module regulates centralspindlin association with the midbody. a** Representative confocal images of HeLa cell midbodies upon depletion of PIPKIγ-i3/i5, synchronization at late cytokinesis, and immunostaining of CIT-K and MKLP1. The dashed line indicates the orientation of the cytokinetic bridge. Scale bar: 3 μm. **b** Relative intensity of CIT-K **c** or MKLP1 dots at the midbody. Normalized data are represented as mean ± SD (*n* = 3 independent experiments). Statistics: two-tailed, one-sample t-test. **d** MKLP1 accumulation at the midbody in knockdown cells stably expressing the indicated PIPKIγ-variants. Normalized data are represented as mean ± SD (*n* = 5 independent experiments; see Supplementary Fig. 5a, for representative images). Statistics: two-tailed one-sample t-test. **e** Representative confocal images of HeLa cell midbodies upon knockdown of PIPKIγ-i3/i5 or of SEPT2, synchronization at late cytokinesis, and immunostaining of MKLP1 and acetylated tubulin. Scale bar: 3 μm. **f** Relative intensity of MKLP1 dots

at the midbody. Normalized data are represented as mean ± SD (*n* = 3 independent experiments). Statistics: two-tailed one-sample t-test. **g** Ultrastructure expansion microscopy (U-ExM) images indicating defects in centralspindlin accumulation at the midbody upon depletion of PIPKIγ-i3/i5. Expanded gels were immunostained for MKLP1, MgcRacGAP, and α-/β-tubulin (note that both tubulins were stained simultaneously and detected with the same secondary antibody to enhance the signal), and imaged on a spinning disk confocal microscope. Representative images (max intensity projections of 21 slices with 1 μm spacing) are shown. Scale bar: 5 μm. **h** Endogenous SEPT2 or **i** endogenous PIPKIγ were immunoprecipitated from lysates of synchronized HeLa cells. The affinity-purified material was separated by SDS-PAGE and analyzed by Western blotting using the indicated antibodies. *$P < 0.05$; **$P < 0.01$. Source data and *P*-values are provided as a Source data file.

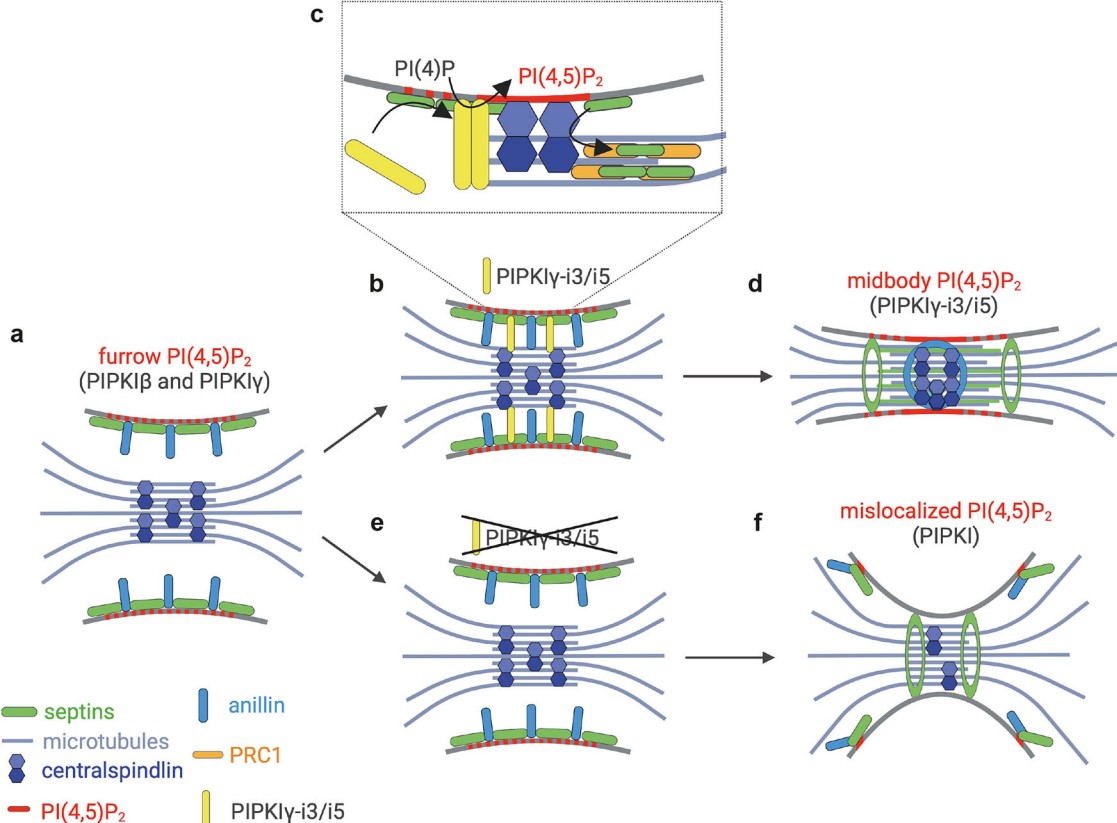

**Fig. 6 | Model summarizing the role of PIPKIγ-i3/i5 during cytokinesis. a** Anillin (turquoise) and septins (green) are enriched at the cleavage furrow, where they act as scaffolds to initiate and sustain actomyosin-mediated constriction. **b** Through interaction with septins, PIPKIγ-i3/i5 (yellow) are recruited to the furrow, where they generate a local pool of PI(4,5)P$_2$ (red). This pool is required for the maintenance of anillin and centralspindlin (blue) at the nascent midbody. **c** The centralspindlin-mediated tethering of the furrow membrane to microtubules possibly facilitates the translocation of septins onto bridge microtubules, and septin

association with PRC1 (orange). **d** Presence of septins on bridge microtubules favors their bundling, while the ICB matures and elongates. **e** Depletion of PIPKIγ-i3/i5 disrupts the localized synthesis of PI(4,5)P$_2$ at the nascent midbody, impairing centralspindlin tethering to the plasma membrane. **f** Mislocalized synthesis of PI(4,5)P$_2$ may contribute to the scattering of anillin and septins. Disorganized septins (and anillin) further affect the maintenance of centralspindlin at the midbody and the maturation of the ICB. (Created with BioRender.com).

bridge. Of note, however, in those cells, mCherry-PIPKIγ-i5 appeared largely excluded from ICBs (Supplementary Fig. 5e). We conclude that stable retention of septin-binding splice variants of PIPKIγ at the ICB requires concomitant presence of anillin, MgcRacGAP, and intact septin filaments.

Taken together, these findings demonstrate that the spatiotemporally controlled recruitment of septin-associated PIPKIγ isoforms to the ICB is essential to maintain centralspindlin at the midbody, via a mechanism that involves the local generation of PI(4,5)P$_2$, which in turn is required for microtubule organization at the ICB.

## Discussion

Almost two decades ago, PI(4,5)P$_2$ was identified as a key factor required for cytokinesis[10], and over time, a steadily increasing number of components of the cytokinetic machinery have been identified that depend on this phosphoinositide for their recruitment to the plasma membrane[9]. Yet, the molecular mechanisms underlying its synthesis at the cleavage furrow and at the maturing ICB have remained elusive. Our study provides important insight into how PIPKIγ splice variants are recruited to the midbody to generate a local pool of PI(4,5)P$_2$ dedicated to maintain centralspindlin (Fig. 6). We demonstrate that a

distinct splice insert (i.e., encoding PIPKIγ-i3 and -i5) promotes the association of PIPKIγ with septins in interphase and in cytokinetic cells. In agreement with a function of PIPKIγ-i3/i5 during cell division, depletion of PIPKIγ causes multinucleation to a similar extent as loss of septins[19,55] and leads to a dramatic scattering of anillin and septins away from the ICB during cytokinesis. Further, we identify centralspindlin as a common binding partner of septins and PIPKIγ, and show that loss of either septins or PIPKIγ impairs centralspindlin accumulation at the midbody.

Based on our observations, we propose the following model for the spatiotemporally controlled synthesis of PI(4,5)P$_2$ by septin-associated isoforms of PIPKIγ during cytokinesis: In cytokinetic cells, anillin-associated septins recruit PIPKIγ-i3/i5 to the constricting cleavage furrow. At this stage, PIPKIγ, likely together with PIPKIβ[16], contributes to cytokinetic progression by fueling PI(4,5)P$_2$ synthesis at the furrow (Fig. 6a). When the cleavage furrow encounters spindle microtubules at the ICB (Fig. 6b) septins and septin-bound PIPKIγ-i3/i5 associate with centralspindlin, to orchestrate the generation of an additional, midbody-close pool of PI(4,5)P$_2$. This pool is essential to retain septins, anillin, and centralspindlin at the nascent midbody (Fig. 6c). Centralspindlin-dependent tethering of microtubules to the furrow membrane then promotes the translocation of septins onto microtubules. This process may be further facilitated by the interaction of septins with PRC1, and serve to foster PRC1-dependent microtubule bundling at the elongating ICB (Fig. 6d). In particular, octameric SEPT9-containing septin complexes that are connected to anillin in a CIN85-dependent manner[45], might confer additional microtubule binding and bundling properties[56]. In the absence of PIPKIγ-i3/i5, PI(4,5)P$_2$ levels at the midbody are reduced, and centralspindlin and anillin anchorage at the midbody are impaired (Fig. 6e). Reduced centralspindlin levels at the midbody, in turn, trigger a microtubule bundling defect, which is accompanied by a loss of the microtubule bundling factor PRC1. Septins, which would associate with bundled microtubules under control conditions upon their release from the plasma membrane, instead associate with PI(4,5)P$_2$-enriched sites in the periphery of the ICB that are likely formed by a compensatory activation of other type I PIPK enzymes. Loss of midbody PI(4,5)P$_2$ also impairs anillin deposition at the ICB. Instead, anillin is mistargeted to septin-enriched sites adjacent to the bridge. As a consequence, microtubules are less stabilized and bundled, and the cytokinetic bridge fails to elongate (Fig. 6f). According to this model, we propose that the assembly of septins, PIPKIγ, and centralspindlin establishes an important checkpoint that arrests cytokinesis progression until the ingressed cleavage furrow has successfully encountered the midbody. Consistently, depletion of septins[19,20], manipulation of local PI(4,5)P$_2$ levels at the midbody[10], expression of anillin mutants deficient in septin binding[12,42], or of membrane-binding-defective MgcRacGAP C1 domain mutants[25] result in similar defects, i.e., furrow instability, eventual retraction of the cleavage furrow, and ultimately multinucleation. Several lines of evidence show that this late function of PIPKIγ strictly requires the lipid kinase activity of PIPKIγ-i3/i5: First, loss of PIPKIγ-i3/i5 causes a prominent reduction in local PI(4,5)P$_2$ levels at the midbody, while, conversely, depletion of the PI(4,5)P$_2$ 5-phosphatase OCRL locally elevates midbody PI(4,5)P$_2$. Second, genetic epistasis experiments show that phenotypes of PIPKIγ-i3/i5 loss can be reversed by co-depletion of OCRL. Third, we demonstrate that PI(4,5)P$_2$ synthesis by active PIPKIγ-i3/i5 is required to rescue defective recruitment and spatiotemporally controlled localization of septins, anillin, and centralspindlin at the nascent midbody of PIPKIγ-i3/i5-depleted cells.

Previous reports based on super-resolution imaging suggested that septins and anillin form ring-like arrays along the ICB to promote its elongation[42]. In support, we find that in kinase-depleted cells, i.e., when anillin and septins are scattered away from the ICB, the acetylated tubulin bridge is significantly shorter. As this defect can only be rescued by wildtype, but not by septin-binding-deficient or catalytically-inactive variants of PIPKIγ, our data implicate septin-dependent synthesis of PI(4,5)P$_2$ by PIPKIγ also in ICB maturation.

To gain insight into the nanoscale organization of microtubules at the ICB, we employed super-resolution expansion microscopy. We demonstrate here that U-ExM is a powerful technique ideally suited to resolve individual structures in crowded environments such as the midbody. Our analyses revealed that microtubules interdigitate at the midbody in wild-type and in PIPKIγ kinase-depleted cells, i.e., independent of local centralspindlin levels. However, depletion of PIPKIγ completely displaced septins from ICB microtubules in regions distal to the midbody. Consistent with the established roles of septins in regulating microtubule organization[18], PIPKIγ kinase-depleted cells displayed significantly reduced levels of acetylated tubulin and PRC1 at the bridge, indicating defects in microtubule stability and bundling, respectively. Future studies will need to address the underlying molecular mechanisms in more detail.

Midbody microtubules are tethered to the plasma membrane by the MgcRacGAP subunit of centralspindlin, which bears an atypical C1 domain with specificity for PI(4,5)P$_2$[25]. Our findings suggest that centralspindlin maintenance anchorage at the midbody is mediated by a tight functional interplay between septins and PI(4,5)P$_2$ synthesized by septin-associated PIPKIγ. Accordingly, we find that depletion of septins, or of PIPKIγ, impede centralspindlin deposition at the midbody to a similar extent. MgcRacGAP´s C1 domain is dispensable for the localization of centralspindlin per se, as shown for variants lacking the C1 domain, or for mutants deficient in PI(4,5)P$_2$-binding[25]. We conclude that its anchorage at the midbody further relies on a septin scaffold that needs to assemble at the ICB. Given that we failed to detect septins at the midbody proper in expanded cell samples (i.e., by U-ExM), we suggest that septins may assist centralspindlin compartmentalization by acting as a diffusion barrier to prevent its premature loss from the midbody rather than serving as a direct centralspindlin anchor. Such a diffusion barrier could also aid the further enrichment of PI(4,5)P$_2$ along the ICB[57].

The motor subunit of centralspindlin, MKLP1, has been shown to associate directly with Arf6 to promote its recruitment to the midbody[58]. Arf6 is a small GTPase required for the completion of cytokinesis[59] and is known to directly activate PIPKIγ[60]. This raises the intriguing possibility that the septin/PIPKIγ-dependent centralspindlin deposition at the midbody acts as a feed-forward mechanism to locally enhance PI(4,5)P$_2$ production and, thereby, foster recruitment of further midbody components, such as exocyst[61,62]. One of the key functions of centralspindlin throughout cytokinesis is the regulation of Rho GTPases, which is exerted through MgcRacGAP[26]. Loss of centralspindlin from the midbody in PIPKIγ-depleted cells might, thus, lead to a misregulation of GTPase activities at the midbody and along the ICB. Consistently, RhoA hyperactivity has been observed in PIPKIγ knockout cells[63]. Hyperactive RhoA, in turn, is known to stimulate the activities of various type I PIPKs[64]. This mechanism could induce PI(4,5)P$_2$ production by other isoforms at aberrant sites, and might explain why loss of PIPKIγ does not result in a complete block of cytokinesis. Future studies will be needed to explore such compensatory mechanisms in detail.

Previous reports suggest important roles of the midbody after completion of cell division[39]. The accumulation of postmitotic midbodies contributes, for instance, to the reprogramming of induced pluripotent stem cells and to the tumorigenicity of cancer cells[65]. In addition, postmitotic midbodies can serve as signaling platforms[66,67]. These functions likely depend on the capability of the midbody to associate with ribonucleoproteins, and with mRNAs encoding proteins involved in cell fate, oncogenesis, and pluripotency[68]. Intriguingly, the formation of such translationally active midbody granules has been shown to depend on MKLP1[68]. Thus, the impaired recruitment of MKLP1 observed in kinase- or septin-depleted cells might further

impact on the downstream signaling of postmitotic midbodies. Future studies will need to reveal potential effects on stemness, tumorigenicity, and proliferation.

## Methods

### Plasmids and siRNAs

For mammalian expression constructs, the coding sequences (CDSs) of human PIPKIγ isoforms 1–5 (PIPKIγ-i1-i5)[50], and of i5 mutants were inserted into pcDNA3.1(+)-based vectors, resulting in the expression of N-terminally mCherry- or HA- tagged proteins. The mutant PIPKIγ-i5 K188A carries the previously described mutation within the kinase core domain, which renders the kinase inactive (Krauss et al., 2006). The mutant deficient in septin binding, PIPKIγ-i5 ΔSB, was generated by mutating Y646 and W647 of human PIPKIγ-i5 into alanine by site-directed mutagenesis (Y646A/W647A). SiRNA-resistant PIPKIγ-i3, -i5, and -i5 mutants were created by introducing four silent mutations within the sequence targeted by the siRNA against PIPKIγ-i3/i5 as follows: 5′-CGACGGCAGATACTGGATT-3′. For viral constructs, the resulting CDSs were subsequently inserted into pLIB-CMV-mCherry-IRES-Puro. For bacterial expression constructs, the CDSs of the tail domains of PIPKIγ-i1-i5 and -i5 ΔSB (aa 451 to end) were inserted into pGEX-4T1, allowing for the expression of N-terminally tagged glutathione-S-transferase (GST) fusion proteins.

His$_6$-PRC1 was a gift from Stephen Royle (Addgene plasmid #69111; http://n2t.net/addgene:69111)[69]. The coding sequence was subcloned into a pcDNA3.1(+)-based vector, resulting in the expression of an N-terminally eGFP-tagged protein.

Most siRNAs used in this study were purchased from Sigma-Aldrich and had 3′-dTdT overhangs. The following siRNAs were used, targeting the human sequences: siOCRL 5′-GAAAGGAUCAGUGUCGAUA-3′[51], siSEPT2 5′- GCCCUUAGAUGUGGCGUUU-3′[70], siSEPT6 5′-CCUGAAGUCUCUCGGACCUAGU-3′[19], siPIPKIγ 5′-GAGGAUCUGCAGCAGAUUA-3′, siPIPKIγ-i5 5′-CAGAAGGGCUUUGGGUAA-3′[36], siPIPKIγ-i3/i5 5′-GGAUGGGAGGUACUGGAUU-3′[36]. On-Target Plus siRNA smart pools (Dharmacon) were used to silence PIPKIα (L-004780-00-0010), PIPKIβ (L-004058-00-0010), MgcRacGAP (L-008650-00-0005), anillin (L-006838-00-0005), or SEPT7 (L-011607-00-0005). The control siRNA used throughout the study had the following sequence: 5′- GUAACUGUCGGCUCGUGGU-3′.

### Generation of antibodies

An antibody specifically recognizing PIPKIγ was raised in rabbits immunized with recombinant His-tagged PIPKIγ (aa451-668)[71]. The same recombinant protein was covalently coupled to CNBr-coupled sepharose and used for affinity purification. An antibody specifically detecting SEPT6 was raised in rabbits immunized with a peptide comprising aa413-327 of SEPT6 (CAGGSQTLKRDKEKKN), and affinity-purified on the same epitope.

### Cell lines

HeLa and HEK-293T cells were obtained from American Type Culture Collection (ATCC), and not used beyond passage 30 from original derivation. The genome-edited NRK49F SEPT2-eGFP knock-in cell line has been described previously[47]. HeLa cells were cultured in Dulbecco`s modified Eagle`s medium (DMEM) containing 1 g/L D-glucose and phenol red (PAN Biotech), supplemented with 10% (vol/vol) heat-inactivated fetal bovine serum (FBS, Gibco), 2 mM L-glutamine (Gibco), 50 μg/mL penicillin-streptomycin (pen-strep, Gibco). Stably transfected HeLa cells were generated by viral transduction, and maintained under selection pressure by additionally supplementing the above-described medium with 1 μg/mL puromycin (Invivogen). HEK-293T cells were cultured in DMEM containing 4,5 g/L D-glucose, phenol red, L-glutamine (Gibco), supplemented with 10% FBS and 50 μg/mL pen-strep. NRK49F SEPT2-eGFP cells were cultured in DMEM containing 4,5 g/L D-glucose (Gibco), supplemented with 10% FBS and 2 mM

L-glutamine. All cell lines were cultured at 37 °C and 5% of $CO_2$, and regularly tested for mycoplasma contamination.

### Generation of an eGFP-SEPT6 knock-in cell line

Endogenous tagging of the SEPT6 N-terminus with eGFP was achieved via the CRISPR/Cas9 technology[72]. In brief, the primers 5′- CACCGC ATCGCTCCTGCGTCCGCCA-3′ and 5′- AAACTGGCGGACGCAGGAGCG ATGC-3′ were annealed and subcloned into px458- pSpCas9(BB)−2A-GFP (Addgene) using the BpiI restriction site to generate a guide RNA. In a second donor vector, the expression cassette was exchanged with the CDS of eGFP inserted between two homology regions (HR) consisting of original genomic sequences ~1000 bp upstream of the SEPT6 ATG (5′HR) and ~1000 bp downstream of the SEPT6 ATG (3′HR). The stop codon of eGFP was exchanged with two codons encoding for a Gly-Ser linker. Design and cloning of the donor vector were performed with the NEBuilder assembly tool and NEBuilder HiFi DNA assembly cloning kit, respectively, according to the manufacturer instructions. HeLa cells were co-transfected with px458- pSpCas9(BB)−2A-GFP containing the guide sequence, and with the donor vector. 72 h later, eGFP-expressing HeLa were sorted into 96-well plates at a density of one cell per well, using a fluorescence-activated single cell sorter (BD FACSAria). Growing colonies were expanded and tested for the expression of eGFP-SEPT6 by automated live cell imaging and Western blotting using anti-SEPT6 and anti-GFP antibodies. The expression of eGFP-SEPT6 in selected clones was further validated by siRNA-mediated depletion of SEPT6, by immunoprecipitation of GFP, and by immunocytochemistry.

### siRNA-mediated gene silencing

To silence PIPKIα, PIPKIβ, PIPKIγ, and PIPKIγ-i3/i5, two rounds of ~48 h knock-down each were performed with 50 nM siRNA, using JetPRIME (Polyplus) as transfection reagent according to the manufacturer´s instructions. For immunocytochemistry (ICC) and Ultrastructure expansion microscopy (U-ExM, see below), cells were seeded on Matrigel (Corning)-coated glass coverslips in a 12-well plate for the second round of knockdown. For live-cell imaging of eGFP-SEPT6, the second round of knockdown was performed in matrigel-coated 8 well glass-bottom slides (ibidi). Cells were analyzed ~48 h after the second round of transfection. Silencing of SEPT2 and SEPT6 was achieved with one round of knockdown for ~48 h with 100 nM siRNA and JetPRIME as transfection reagent according to the manufacturer´s instructions.

For the co-depletion of PIPKIγ-i3/i5 and OCRL, two rounds of knockdown were performed, with 50 nM siPIPKIγ-i3/i5 + 50 nM siOCRL. In the same experiment, the single depletions of PIPKIγ-i3/i5 and OCRL were performed with 50 nM targeting siRNA + 50 nM siControl, while control cells were treated with 100 nM siControl at each round.

### Analysis of the expression of PIPKIγ isoforms

RT-PCRs were performed as described previously[73]. Briefly, 1 μg of RNA was used with isoform-specific reverse primers for the RT-reaction, and the subsequent PCR was performed with a $^{32}$P-labeled forward primer. Products were separated by denaturing PAGE and quantified using a Phosphoimager (Typhoon 9200, GE Healthcare) and Image-QuantTL software. The sequences of the primers used for the initial PCRs and the expected sizes of the amplicons were as follows: Common PIPKIγ forward: GCGCCCGCCACCGACATCTAC; PIPKIγ-i1-3 reverse: CATCTCCCGAGCTCTGGGCCTC (i1 = 125 nt, i2 = 210 nt, i3 = 290 nt); PIPKIγ-i4 reverse: GAGACCAGGACGCGCACAAACCAG (i4 = 154 nt); PIPKIγ-i5 reverse: CAGACACTGAGCTTCCGGCCGG (v5 = 195 nt). For isoforms i3 and i5, we further optimized the PCR by using a reverse primer that amplifies a region common to the i3 and i5 isoforms (v3_5rev: AGTCCCCGAGGCGCTC) with the common forward primer (amplicon size 100 nt). To quantify specific knockdown of i3/i5

isoforms, we normalized the i3/i5 product to i1, which was the main isoform.

## Plasmid overexpression

Transfection of HeLa was performed with JetPRIME (Polyplus), according to the manufacturer´s instructions. The medium was exchanged after 6 h of transfection, and the cells were processed after 24 h of transfection. For the overexpression of PIPKIγ isoforms, cells were seeded on matrigel-coated glass coverslips in a 12-well plate, and transfected with 0.5 μg of DNA per well.

Transfection of NRK49F SEPT2-eGFP was performed with Lipofectamine 3000 (Thermo-Fisher), following the manufacturer´s instructions. Cells were seeded on glass coverslips in a 6-well plate, transfected on the following day, and used for experiments 48 h post-transfection. For immunoprecipitation assays or for virus production, HEK-293T cells were transfected using calcium phosphate. In brief, DNA was mixed with 0.12 M $CaCl_2$ in 0,1 × TE buffer (1 mM TRIS, 0.1 mM EDTA, pH 8.0) and incubated for 5 min at room temperature. The same volume of 2X HBS (280 mM NaCl, 10 mM KCl, 1.5 mM Na2HPO4, 12 mM dextrose, 50 mM HEPES, pH 7.0) was added while stirring. After 20 min of incubation at room temperature, the DNA solution (1 mL for a 10 cm dish) was added to the cells.

## Cytochalasin D treatment and phalloidin staining in NRK49F SEPT2-eGFP

Cells were incubated for 30 min in full medium containing 5 μM cytochalasin D. After a quick rinse with PHEM buffer (60 mM PIPES, 25 mM HEPES, 2 mM $MgCl_2$, 10 mM EGTA, pH 6.9), cells were fixed with 4 % PFA in PHEM buffer at 37 °C for 15 min. Excess of PFA was quenched with 50 mM $NH_4Cl$ in PHEM buffer for 10 min on a shaking platform. Cells were washed three times for 5 min on a shaking platform, and stored for imaging, or processed for phalloidin staining.

For phalloidin staining, cells were permeabilized with 0.3% Triton-X 100 in PHEM buffer for 5 min. Next, samples were blocked with 4% horse serum and 1% BSA in PHEM buffer for 1 h. Coverslips were incubated with 1 μM AlexaFlour-647-coupled phalloidin (Thermo-Fisher) in 1% BSA/PHEM buffer for 30 min, and then washed three times for 10 min in PHEM buffer on a shaking platform. Coverslips were mounted in FluoromountG (Invitrogen).

## Immunocytochemistry

HeLa cells seeded on matrigel-coated glass coverslips were fixed with 4% PFA/4% sucrose, or with 2% PFA/2% sucrose (for septin stainings) in PBS for 15 min at RT, and subsequently washed three times with PBS. Cells were permeabilized with washing buffer (20 mM Hepes pH 7,2, 150 mM NaCl, 0.1% Triton X-100) for 15 min, and then blocked with goat serum dilution buffer (GSDB: 20 mM Hepes pH7.2, 150 mM NaCl, 0.1% Triton X-100, 10 % goat serum) for 20 min. Incubation with primary antibodies, diluted in GSDB, was carried out at RT for 1 h, and the excess of antibody was removed by three washes of 10 min each with washing buffer. Cells were subsequently incubated with AlexaFluor-conjugated secondary antibodies diluted in GSDB for 1 h at RT. After three washes in washing buffer, coverslips were incubated for 5 min with 1 μg/mL of DAPI in PBS, and mounted on microscope glass slides with Immu-Mount (Thermo-Fisher). For the staining of anillin, goat serum was replaced by donkey serum throughout the protocol.

Plasmalemmal $PI(4,5)P_2$ was visualized using purified, recombinant pleckstrin homology (PH) domain of PLC-δ1. Cells were fixed with 2% PFA, 2% sucrose + 1% glutaraldehyde (GA) in PBS for 20 min at RT. Excess of fixative was quenched by three washes with 50 mM $NH_4Cl$ in PBS. Permeabilization and blocking were performed concomitantly with a first round of incubation with 0.25 μg/mL of $His_6$-tagged eGFP-PH-PLCδ1 in 0.5% saponin, 1% BSA in PBS for 30 min at RT. Subsequently, cells were incubated (without washing in between) again with 0.25 μg/mL of $His_6$-tagged eGFP-PH-PLCδ1 diluted in 1% BSA in PBS for

30 min at RT. Subsequently, coverslips were washed three times with PBS, and incubated with primary antibody (rabbit-anti-GFP, Abcam) in 1% BSA, 10% GS in PBS for 1 h at RT. Excess of antibody was removed by three washes with PBS. The secondary antibody (AlexaFluor488-coupled goat-anti-rabbit) was diluted in 1% BSA, 10% GS in PBS and incubated at RT for 30 min. Excess of antibody was removed by four washes with PBS, supplemented with 1 μg/mL of DAPI for the final wash before mounting. To visualize $PI(4,5)P_2$ at the midbody by antibody labeling (mouse-anti-$PI(4,5)P_2$, Echelon), fixed cells were permeabilized in 0,5% saponin, 1%BSA in PBS for 30 min at RT. Coverslips were incubated with primary antibodies diluted in 1% BSA, 10% GS in PBS for 2 h at RT. Excess of antibodies was removed by three washes in PBS, and coverslips were incubated with AlexaFluor-coupled secondary antibodies diluted in 1% BSA, 10% GS in PBS for 1 h at RT. Coverslips were subsequently washed three times with PBS, once with PBS supplemented with 1 μg/mL of DAPI, and finally mounted.

## Cell synchronization

For microscopy-based assays, 2 mM thymidine was applied in full medium to stall the cells at S-phase. 24 h later, thymidine was removed by four washes of 1 min with PBS, and cells were incubated for 7,5 h in fresh medium. Then, fresh medium supplemented with 20 ng/mL of nocodazole was added. After 4 h, cells were carefully washed four times for 1 min with full medium pre-warmed at 37 °C, and subsequently allowed to proceed to cytokinesis by incubation for 1.5 h in full medium, before fixation. For live cell imaging, the nocodazole block was omitted.

For immunoprecipitation, the nocodazole block was applied overnight by supplementing the medium with 40 ng/mL of nocodazole. The following day, cells were collected by mitotic shake-off and washed four times in full medium by centrifugation at $300 \times g$ for 5 min. Cells were then re-plated in full medium and allowed to proceed to cytokinesis for 90 min. Cells were collected by gentle shake-off and harvested by centrifugation at $300 \times g$ for 5 min. Before lysis, cells were washed once in PBS. Cells stalled at the S-phase for 48 h served as controls and were harvested by trypsinization.

## Microscopy and image analysis

Immunostained HeLa cells were routinely imaged with a Zeiss confocal spinning disk microscope (Yokogawa CSU22, Hamamatsu EMCCD camera), using a 60× immersion oil objective (1.4 NA). At least 12 images were acquired per condition, per experiment. For each image of cytokinetic cells, a stack of 21 pictures within the z-plane (z-stack), with a spacing of 0.2 μm was acquired. Cells stained with purified $His_6$-tagged eGFP-PH-PLCδ1 domain (Supplementary Fig. 1e), antibodies recognizing $PI(4,5)P_2$ (Fig. 3d), and knock-in eGFP-SEPT6 cells (Supplementary Fig. 4c) were imaged within a single z-plane by focusing on the plasma membrane, or on prominent septin filaments, respectively. For the experiment depicted in Fig. 1 and Supplementary Fig. 1b, semi-automated epi-fluorescent imaging was conducted with a Nikon Eclipse Ti microscope (illumination: CoolLED, pE4000, prime95B sCMOS camera) operated by NIS-Elements software, using a 20× air objective (0.75 NA). Tile scans of 1.3 mm² were produced by stitching together images automatically acquired around a chosen point. Four tile scans were acquired per condition, per experiment. Epi-fluorescent pictures of HeLa cells transfected with PIPKIγ isoforms (Supplementary Fig. 2c) were acquired with the same microscope, using a 40× immersion oil objective (1.3 NA). Live cell imaging of eGFP-SEPT6 cells was performed on a spinning disk Nikon Eclipse Ti microscope (Yokogawa CSU-X1 and EMCCD Camera), operated by NIS-Elements software, with a 40x air objective (0. 95 NA). Imaging was initiated 7, 5 h after thymidine release, and was carried out overnight, with a frame rate of 10 min. Cells were kept in full medium at 37 °C and 5% $CO_2$, and pictures were acquired within a single z-plane that was set at the beginning by focusing on septin fibers of pre-mitotic cells, and kept by

an autofocus system. NRK49F SEPT2-eGFP cells and gels for U-ExM were imaged with an Olympus spinning disk microscope (Yokogawa CSU-X1, Hamamatsu C11440 camera), using a 60x immersion oil objective (1.42 NA). Cells depicted in Fig. 2d, e were imaged within a single z-plane by focusing on septin fibers. For cells depicted in Fig. 2f, a z-stack with a spacing of 0.3 μm was acquired. For U-ExM, 20 images were acquired per condition, per experiment. For each image, a z-stack of 21 layers with a spacing of 1μm was acquired. For imaging of SiR-tubulin, cells were seeded into Matrigel-coated Ibidi 8-well glass-bottom dishes and incubated with 100 nM SiR-tubulin (Spirochrome) in full media overnight. Imaging was performed in a live-cell incubator with 5% CO2 (OkoLab) heated at 37 °C, on a NikonTiE2 (Nikon) equipped with a confocal spinning disk unit (CSU-W1, Yokogawa) with a PL APO λ 40x/0.95 NA air objective without additional magnification. The microscope was equipped with two sCMOS cameras (pco.edge, 4.2bi, 6.5 μm/pixel, 2048 × 2048 pixel) and controlled with NIS-Elements software (Nikon). Images were taken with the following sequential fluorophore settings: DAPI (Ex.: 405 nm; Em.: 420–460 nm), EGFP/AF488 (Ex.: 488 nm; Em.: 500–550 nm), JFX554 (Ex.: 561 nm; Em.: 574–626 nm), AF647 (Ex.: 638 nm; Em.: 670–746 nm). Live imaging was performed in the AF647 channel with 10 min frame interval, and videos were analyzed by measuring the time from initiation of mitosis (marked by cell rounding) until completion of furrow ingression (marked by the appearance of a thin microtubule bridge), and thereafter until microtubule cut (marked by a sudden retraction of the microtubule bundle from the ICB into one of the daughter cells).

Image processing and analysis were performed with the open-source software Fiji (ImageJ).

Quantifications depicted in Fig. 1b–e were performed by identifying and manually counting mitotic cells displaying a mitotic spindle, or a cytokinetic bridge, as schematized in Fig. 1a. This number was then divided by the total number of nuclei, identified via a macro and serving as the total number of cells in the picture.

The qualitative assessment of anillin at the ICB was conducted on maximum intensity projections of z-stacks. For measuring the intensity of the acetylated tubulin bridge, average intensity projections of the z-stacks were generated, and the acetylated tubulin channel was segmented in order to obtain regions of interest (ROI) outlining the acetylated tubulin bridges. Subsequently, the fluorescence intensity of acetylated tubulin was measured as integrated density within the obtained ROI on average intensity z-projections, after background subtraction. The intensity of PRC1 bridges, MKLP1, and MgcRacGAP dots was measured the same way, in regular confocal or expanded samples. The length of the acetylated tubulin bridges was measured as the max Feret diameter of the ROI outlining the acetylated tubulin bridges. Septin enrichment at the acetylated tubulin bridge was determined by dividing the intensity of SEPT2 in a ROI outlining the acetylated tubulin bridge by the intensity of SEPT2 in a ROI outlining the whole dividing cell (drawn by hand); here, as well, measurements were carried out on average intensity z-projections, after background subtraction. In quantifications depicted in Supplementary Fig. 1f, the intensity of PI(4,5)P$_2$ per cell area was measured as the mean gray value of the eGFP-PH-PLCδ1 fluorescence in the ROI outlining single cells (drawn by hand), after background subtraction. Intensity line scan analysis, as depicted in Fig. 2g, was performed on maximum intensity projections of the z-stacks. Midbody PI(4,5)P$_2$ and CIT-K levels, as depicted in Fig. 3e, f were determined by placing an equally sized ROI of 3 μm in diameter on the CIT-K signal at the midbody. Intensities were measured upon background subtraction.

For all displayed images, brightness and contrast were adjusted equally for different conditions, unless otherwise indicated.

## Ultrastructure expansion microscopy (U-ExM)
Samples were processed using the U-ExM method[53]. Abbreviations used: FA (Formaldehyde), AA (Acrylamide), BIS (N,N'-Methylenebisacrylamide),

TEMED (N,N,N',N'-Tetramethylethylenediamine), APS (Ammonium persulfate), SA (Sodium acrylate). HeLa cells (seeded on 18 mm matrigel-coated coverslips) were rinsed with PHEM buffer (60 mM PIPES, 25 mM HEPES, 2 mM MgCl$_2$, 10 mM EGTA, pH 6.9), then fixed in 4% PFA, 0.1% Triton X-100 in PHEM at 37 °C for 15 min. Fixed coverslips were rinsed in PHEM buffer again and transferred into 1 mL of anchoring solution (0.7% FA, 1% AA in PBS). Samples were incubated for 16 h at RT. U-ExM monomer solution (19% (wt/wt) SA, 10% (wt/wt) AA, 0.1% (wt/wt) BIS in PBS) was prepared 1 day before the gelation and stored at −20 °C until use. 90 μL of monomer solution was mixed with 5 μL of 10% TEMED and 5 μL of 10% APS (0.5% APS and TEMED in the final monomer solution) and placed on Parafilm in a drop in a pre-cooled humid chamber. The coverslips were placed on the drops facing downwards and incubated on ice for 5 min. For gelation, coverslips were incubated at 37 °C for 1 h in the humid chamber. Gels (still on coverslips) were placed in 2 mL denaturing buffer (200 mM SDS, 200 mM NaCl, 50 mM Tris, pH9) in a six-well plate for 15 min at RT with agitation. For denaturation, gels were transferred into 15 ml centrifuge tubes filled with 3 mL of denaturation buffer and incubated at 95 °C for 1 h. Subsequently, the gels were washed several times with MilliQ water and then expanded in 10 cm dishes in MilliQ water. The water was exchanged twice after 30 min. Finally, gels were left to expand in water overnight at 4 °C.

Gels were measured with a caliper the following day to calculate the expansion factor. Gels were cut into 2 × 2 cm squares, and the squares were incubated twice in PBS for 15 min. The shrunken gels were stained with antibodies diluted in 2% BSA/PBS. To this end, each gel piece was placed in a 12-well plate, covered with 500 μL of antibody solution, and incubated for 3 h at 37 °C and at 120 rpm. Gels were then washed three times for 20 min with PBS-T (PBS supplemented with 0.1% Tween20) in a 6-well plate with gentle agitation. Secondary antibody incubation was executed as indicated above for 2.5 h. Gels were then washed three times for 20 min with PBS-T in a 6-well plate under gentle agitation. Final expansion was done in MilliQ water in 10 cm dishes, exchanging the water at least two times after 30 min, and letting the expansion plateau by an overnight incubation at 4 °C. For imaging, gels were mounted in imaging dishes (Zell-Kontakt) with 2% agarose gels. To calculate the scale, the pixel size of the camera was divided by the measured gel expansion factor (~4,2).

## Cell lysates
Cells were trypsinized, washed with PBS, and lysed for 15 min on ice in lysis buffer (20 mM HEPES, 100 mM KCl, 2 mM MgCl$_2$, 1% Triton X-100, pH 7,4) supplemented with protease inhibitor cocktail (Sigma-Aldrich) and 1 mM PMSF. Lysates were cleared by centrifugation at 17,000 g for 15 min at 4 °C, and the protein concentration was determined with Bradford reagent (Sigma-Aldrich). Lysates were subsequently denatured in Laemmli sample buffer for 5 min at 95 °C. For each sample, 10–30 μg of protein were resolved by SDS-PAGE and analyzed by western blot (see below).

## Immunoprecipitation assay
HEK-293T cells seeded in 10 cm petri dishes were transfected at 70% confluency with 5 μg of plasmid encoding mCherry-PIPKIγ or mCherry alone and 5 μg of myc-tagged SEPT7 or SEPT9, using the calcium phosphate method. Twenty-four hours later, cells were trypsinized, washed once with PBS, and lysed at 4 °C for 15 min in lysis buffer (20 mM HEPES, 100 mM KCl, 2 mM MgCl$_2$, 1% Triton X-100, pH 7,4) supplemented with protease inhibitor cocktail (Sigma-Aldrich), 1 mM PMSF, and phosphatase inhibitors cocktails 2 and 3 (Sigma-Aldrich). Lysates were cleared by centrifugation at 17,000 × g for 15 min. 0.6 mL of the resulting supernatant (containing 2–4 mg of proteins) was supplemented with 15 μL of RFP-Trap magnetic particles (Chromo-Tek), and incubated for 2.5 h at 4 °C on a rotating wheel. Beads were washed three times with lysis buffer and a fourth time with lysis buffer

without detergent. Proteins bound to the beads were eluted by boiling in 60 μL of Laemmli sample buffer for 5 min, resolved by SDS-PAGE, and analyzed via immunoblot.

For immunoprecipitation experiments cell lysates derived from HeLa cells arrested in S-phase by thymidine treatment (obtained upon trypsinization), or from HeLa cells in cytokinesis (obtained by gentle pipetting) were incubated with 5 μg of rabbit-anti-human control, of rabbit-anti-SEPT2, or of rabbit-anti-PIPKIγ antibodies, bound to protein A/G magnetic beads (Pierce™) at 4 °C for 3–4 h on a rotating wheel. Beads were washed four times in lysis buffer (20 mM HEPES, 100 mM KCl, 2 mM MgCl₂, 1% Triton X-100, pH 7,4) and once in lysis buffer devoid of detergent. Retained proteins were released by boiling in 90 μL of Laemmli sample buffer for 5 min, resolved by SDS-PAGE, and analyzed by Western blotting.

## GST pulldown assay

Expression and purification of GST-tagged PIPKIγ tails (or of GST alone, used as control) was performed essentially as described previously[71]. All purification steps were carried out on ice. Briefly, the pellet of 250 mL expression culture was resuspended in 35 mL of PBS, sonicated, supplemented with 150 U of benzonase (Sigma-Aldrich), incubated for 10 min, supplemented with 1% Triton-X-100, sonicated, and incubated again for 10 min. The bacterial lysate was cleared by centrifugation at 50,000 × $g$ for 15 min. The supernatant was incubated with 500 μl of GST-Bind Resin™ (EMD Millipore), and incubated for 2 h by end-over-end rotation. The resin was washed three times with 20 mL of PBS, and finally resuspended in 700 μl of PBS. Quantity and integrity of the purified fusion protein were determined by SDS-PAGE, followed by Coomassie Blue.

For pulldown experiments, mouse brain extracts were obtained by homogenizing four brains in homogenization buffer (20 mM HEPES, 320 mM sucrose, pH 7.5) containing complete EDTA-free protease inhibitor cocktail (Roche). The homogenate was centrifuged at 1000 × $g$ for 15 min at 4 °C. The supernatant was recovered, supplemented with 1% Triton X-100, 100 mM KCl, 2 mM MgCl2, and incubated on ice for 10 min. The lysate was cleared by centrifugation at 17,000 × $g$ for 15 min, and subsequently at 178,000 × $g$ for 15 min (4 °C). The supernatant was recovered and used at a concentration of 14 mg protein/mL. Pulldown experiments were performed by incubating 1 mL of protein extract with 70 μg of GST-fusion protein or of GST for 3 h at 4 °C by end-over-end rotation. The samples were subsequently washed four times with buffer containing 20 mM HEPES, 100 mM KCl, 2 mM MgCl2, 1% Triton X-100, pH 7.4, and once in the same buffer without detergent. Proteins were eluted from the beads by boiling for 5 min in 100 μL of Laemmli buffer, and analyzed by Western blotting. Chemiluminescent signals were detected using the Chemie-Doc MP Imaging System (BioRad) controlled by the Image Lab 6.0.1 software. Fluorescent signals were detected using the LI-COR Odyssey Fc imaging system controlled by the Image Studio software.

## Statistics and reproducibility

All quantitative data were derived from at least three independent experiments and are presented as means ± standard deviation (SD). GraphPad Prism version 9.2 software was used for statistical analysis. Unpaired two-tailed t-test was applied to compare two groups. One-way ANOVA followed by Dunnett´s or Tukey´s multiple comparison test was used to compare more than one experimental group to a control group. When the control group was set to 1 by normalization, one sample two-tailed t-test was applied for comparing one or more experimental groups to the control. The level of significance is indicated in the figures by asterisks (*$P < 0.05$; **$P < 0.01$; ***$P < 0.001$; ****$P < 0.0001$), and detailed in the Source Data files as exact $P$ value.

Affinity purification experiments, as shown in Fig. 2b were performed three independent times with PIPKIγ-i2,-i3,-i5, twice including also PIPKIγ-i5 ΔSB, and yielded similar results. Immunoprecipitation experiments, as illustrated in Figs. 2c and 5h, i, were repeated several times with similar results, but not in all experiments all potential interaction partners were analyzed, and not in all experiments lysates of HeLa cells in S-phase were included. In summary, in immunoprecipitates of PIPKIγ, septins were detected in five independent experiments, CIT-K in seven experiments, and centralspindlin in five experiments. In immunoprecipitates of SEPT2, centralspindlin was detected in six independent experiments, anillin in seven experiments, PRC1 in nine experiments, and PIPKIγ in two experiments. Two independent experiments were performed to analyze the co-localization of SEPT2-eGFP with F-actin in genome-edited NRK49F cells, and yielded similar observations to those depicted in Fig. 2d. Co-localization of PIPKIγ-variants with SEPT2-eGFP as was assessed in four independent experiments, with a similar outcome as depicted in Fig. 2e. Results as shown in Fig. 2f were reproduced in three independent experiments. The localization of mCherry-PIPKIγ-i5 was correlated with SEPT2 and anillin in three independent experiments, with similar results as depicted in Fig. 3a. Please see Source Data for additional information on the reproducibility of results shown in Supplementary Figs.

## Reporting summary

Further information on research design is available in the Nature Portfolio Reporting Summary linked to this article.

## Data availability

The authors declare that all relevant information supporting the findings of this study are available within the paper and its supplementary files, or are available from the corresponding authors. Source data are provided with this paper.

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

## Acknowledgements

This work was supported by the German Research Foundation (DFG) within CRC 958 (A11 to M.K. and H.E.; Z02 to H.E.; A01 to V.H.; A21 to F.H.) and TRR 186 (A08 to V.H.). We further thank Heike Stephanowitz and Prof. Fan Liu (FMP Berlin) for mass-spectrometric analyses, and Delia Löwe for help with cell synchronization. We additionally thank Akin Sesver for acquiring initial data with the eGFP-SEPT6 cell line.

## Author contributions

G.R., S.R., N.J., and N.H. carried out the experiments, C.S., M.L., and H.E. helped with image analysis, F.H. determined mRNA levels upon knock-downs of PIPKIγ isoforms. M.K. designed the project and, together with V.H., supervised experiments and wrote the manuscript.

## Funding

## Competing interests

The authors declare no competing interests.
