## [Transparent Peer Review file · Nature Communications]

Local PI(4,5)P2 synthesis by septin-associated PIPK1 γ isoforms controls centralspindlin association with the midbody during cytokinesis

Corresponding Author: Dr Michael Krauss

Version 0:

Reviewer comments:

Reviewer #1

(Remarks to the Author)

Russo et al. present how PIPK1 γ , an enzyme synthesizing PI(4,5)P2, interacts with partners at the intercellular bridge and midbody during cytokinesis. The results demonstrate that the isoform PIPK1 γ has a key role at the midbody as compared with other isoforms α and β . The interplay between key proteins (septin, anillin, centralspindlin) at the midbody and PIPK1 γ is described to propose a model where the synthesis of PI(4,5)P2 tunes the organization of the midbody throughout cytokinesis. Russo et al. have done some interesting work to understand the mechanisms occurring during cytokinesis. However, some control experiments would be welcome to make their points more convincing. Hence, I would recommend some revisions before the manuscript can be accepted in Nature Communications.

Major revisions

- In most experiments, PIPK1 γ is depleted, while the expression, localization of possible partners are assayed. Conversely, depleting septins, anillin, or centralspindlin and describing the expression level and localization of PIPK1 γ would be key to actually understand how PIPK1 γ is controlled and dependent on those key partners. Besides, the authors state, at least twice, their assays demonstrate a "septin-dependent recruitment of PIPK1 γ " while they actually describe a PIPK1 γ -dependent recruitment of septins.
- Testing the role of PIPK1 on myosin II recruitment would be valuable as well given myosin is involved in cytokinesis.
- To demonstrate synergistic roles of the different PIPK1, particularly on the distribution of PI(4,5)P2, depletions of combinations of the isoforms would be worthwhile. Have they authors attempted to deplete combinations of 2 or all 3 simultaneously?
- On Figure 1.h, upon depletion of PIPK1 γ , septins can still be visualized bound to the plasma membrane as well as within the midbody. Was this a frequent observation? Can the authors comment on this? Besides, depletion of PIPK1 β seems to also disrupt septin localization given that septins can be seen as "clusters" or point-like densities. Can the authors comment on this?
- While using expansion microscopy septins seem to localize and organize as rings on both sides of the midbody, this localization is not seen using conventional microscopy. Instead septins seem to colocalize with microtubules in a bundle-like organization. Why would that be different?
- From the literature, it is considered that anillin would recruit septins. However, in this study, the authors conclude that septins would promote anillin recruitment through PIPK1 at the midbody. Can the authors comment on this observation?
- Figure 4: Depletion of PIPK1 induce mis-localization of septins and disruption, to some extent, of the microtubule spindle. Is it a direct effect or does it occur through a PRC1 dependent effect? Can the authors possibly test or at least comment on any direct interaction between PRC1 and septins, for instance? Besides, did the authors consider testing how microtubule disruption would affect septin and PIPK1 distribution in the cell?

Minor revisions

- In different figures, the displayed images are too small (see, for instance, Figures 1.f, 3.d, 3.g, Supl. 1.b, Supl. 4.d, 4.e...). Enlargements would thus be welcome for the evidences to be more convincing.
- It would be worthwhile to show images (in supplementary) corresponding to the phenotypes described in Figure 1.d and 1.e (multipolar spindles and bridges, respectively).
- Can the authors comment on the effect of Thymidine, used for cell synchronization? Can it affect the ultrastructure, stability

of the midbody and/or the recruitment of specific proteins?

-P7, l196: The authors should specify that the affinity purification was performed from a mouse brain lysate.

-Figure 2e.: Some of the cells are not labelled with m-Cherry-PIP $\text{K}\text{I}\gamma$ while a septin signal is visible. Can the authors give an explanation?

-Figure 2.f: Did the authors perform the same experiment testing for the i3 isoform? Did they obtain the same phenotype as for i5 mutant?

-Supplementary Fig.3: The numbering for the different panels differs from the font/numbering used for all other figures. This should be homogenized.

-Page 9, top (l 246-253): The described result is not displayed or it is unclear ?

-Figure 3.d: On the siOCRL panel, the signal from citron kinase seems much higher than in the panels above and below. However the quantification (3.f) indicates a rather constant intensity depending on the conditions. Can the authors comment or correct accordingly?

-Supplementary fig. 4.g: The septin pattern observed for i5 WT looks different from others and septins seem to localize in the midbody as well. Can the authors comment on this?

-Figure 5.k: The signals for MKLP1 and septins from an IP via PIP $\text{K}\text{I}\gamma$ are rather weak. Can the authors comment on this? Testing for centralspindlin, have the authors also assayed RacGap1?

Reviewer #2

(Remarks to the Author)

The manuscript submitted by Russo and colleagues addresses the question of how PIP 2 is generated at the intercellular bridge during mammalian cytokinesis and what effect it has on cytokinetic progression. They report that coupling of the PI-5-P kinase PIP $\text{K}\text{I}\gamma$ with septins is required to stabilize the centralspindlin complex at the midbody/Flemming body in late telophase. This study addresses an important and unresolved question in the field and is thus of significant interest to cell division researchers. Additionally, it reveals an interesting coupling of the septin cytoskeleton to lipid synthesis. However, a number of clarifications are required.

Major points:

1. A major deficiency in the manuscript is a lack of rescues for siRNA-mediated depletions in multinucleate experiments described in Figures 1 and Supp Figure 3D. Given that the phenotypes are small in absolute terms, these rescues are particularly important to establish that the relatively small increases observed for cytokinesis failure are not due to off-target effects. While the rescues shown in Fig 3i are appreciated, they are not direct evidence of a rescue of cytokinesis failure. Rescues would preferably be undertaken with strategies that avoid over-expression.

2. The phenotype induced by PIP $\text{K}\text{I}\gamma$ disruption (either by siRNA alone or with expression of mutants) requires more defining. If PIP KI is important for telophase progression, is there a measurable delay in telophase? Does abscission take longer? Some timing experiments could help clarify

3. Some terms used in the manuscript are unclear and ambiguous. The common one of 'compact' to describe anillin localization is unclear and does not reflect what can be seen in the images. (Fig 1,3). Nor does this reflect previous studies that defined anillin dynamics in the intercellular bridge (El Amine et al. 2013, Renshaw et al. 2014). These two studies characterized in detail anillin dynamics and define common organizational states that reflect different phases of intercellular bridge assembly. Assessing if and/or when these previously defined patterns of anillin localization are disrupted by the various perturbations performed in this study will make comparisons across studies much easier.

4. Related to point 3, in many cases the micrographs being compared appear to be at different stages of intercellular bridge development (i.e. that the authors are comparing two ICBs of differing ages). This occurs frequently in Figs 3, S3 and 4. Including tubulin staining to the experiments would help clarify these issues as the degree of microtubule compaction has been used to assess the intercellular bridge age thereby allowing more direct "apples to apples" comparisons. This is a very important point with respect to the conclusions made and should be rigorously addressed.

Minor points:

A. In the introduction the Piekny and Maddox review is extensively cited. While this is a good review it is 15 years old and much has been learnt subsequently about anillin, who it binds to and how it functions. It would be useful to newcomers to the field if more up to date references about specific points were used.

B. The biology of septins needs a more thorough treatment, again with up to date references. While it is said that septins hetero-oligomerize, the hexamer/octamer distinction is bound to be confusing to non-experts and should be better developed. In addition, in the results section (line 170) the authors state that septins bind to the C-term of anillin. While this is the conclusion drawn from this 2000 paper, subsequent work suggests a more complex relationship. This combined with a better discussion of hexamers and octamers would bring the paper up to date and allow a subsequent more detailed discussion (and perhaps experimental plan) of the underlying mechanism at play.

C. In the results, a simple statement that cells were synchronized by double thy + nocodazole treatment should be added to the main text to aid the reader.

D. Experiments demonstrating the localization of anillin and septins in PIPKly-depleted cells across the stages of cytokinesis (as shown for control cells in Fig 3a) would be helpful. As the authors state, anillin and septin recruitment to the equatorial membrane prior to furrow ingression are known to be dependent on PI(4,5)P2. However, the manuscript only shows disrupted telophase localization of anillin in knockdown/mutant conditions. This may result from deficiencies in recruitment to the furrow/ICB in anaphase-early telophase as opposed to the action of PIPKly later in telophase.

E. What effect does knockdown of PIPK α & β have on PI(4,5)P2 levels in anaphase and telophase cells? The results of Fig 3 would suggest that PIPKly is responsible for the majority of PI(4,5)P2 present in cytokinetic cells. Is the remaining PI(4,5)P2 is sufficient to promote initial anillin/septin ICB recruitment?

F. Fig 4: Measurements comparing the width of ICB bridges across conditions should be performed. If as the authors argue, depletion of PIPKly γ 3-5 perturbs the development of the ICB such that its microtubules do not compact as much as control, then the width of acetylated tubulin staining will better bolster this point than measurements of ICB length.

G. Fig5hi: I do not think it's appropriate to measure fluorescence intensities across different expanded samples unless steps were taken to ensure some normalization of expansion factors and a comparison to some constant. The obvious differences in intensity presented in panel g are more than sufficient to support the authors' claims.

H. In S3G: the i5 Δ supSB panel contains a cell that appears to be at an earlier stage of cytokinesis compared to the other cells in this panel (based on less compaction of tubulin staining). It further also looks to be multinucleated. Is this the norm? If yes this would have implications in interpretation.

I line 76 PI(4,5)2 should be PI(4,5)P2

Reviewer #3

(Remarks to the Author)

This manuscript studies the spatiotemporal control of PI(4,5)P2 during cytokinesis and reports a key role for two splice isoforms of PIPKlgamma. The authors demonstrate that PIPKlg is required for cytokinesis progression and for the organization of anillin and septins at the intracellular bridge (Figure 1). They furthermore identify two splice isoforms (i3 and i5) of PIPKlg that can interact with septins (Figure 2) and demonstrate that these are critical for proper anillin accumulation and PI(4,5)P2 synthesis near the midbody (Figure 3), as well as for the recruitment of septin to the intracellular bridge (Figure 4) and the association of the centralspindlin complex to the midbody (Figure 5). Based on these results, the authors propose a model in which septins recruit PIPKg to the cleavage furrow and generates local PI(4,5)P2, whereas later on septin-bound PIPKg create even more local PI(4,5)P2 that contributes to septin retention and subsequent microtubule stabilization. Overall, the data is convincing and the writing is clear. Nonetheless, I do have a number of comments that need to be properly addressed to make the manuscript ready for publication.

1. I found the authors conclusions on the interdependence of septins and PIPKlg unclear. Initially, the authors postulate that septin recruits PIPKlg to the midzone, whereas later on they suggest that PIPKlg is key for proper localization of SEPT6, including its localization to microtubules. The text and cartoons should be improved to clarify how the authors think about this interdependence.

2. In addition, the whole interplay between septins, PIPKlg and microtubule stabilization is not well developed and therefore not very convincing. How does production of PI(4,5)P2 helps to enrich septins on microtubules?

3. The title and discussion put a lot of emphasis on nanoscale synthesis, but it is unclear if this means anything else than localized, restricted synthesis. Of course, if an individual protein modifies lipids this can always be considered nanoscale synthesis, but the current formulations are raising the expectations that the authors have discovered nanosized membrane regions that carry specific modification. This is not at all reflected in the presented data.

Minor comments

4. Figure S3 – panel labels are inconsistent with all other figures

5. Figure 2f – The merge images of SEPT2 and PIPKg are confusing and would be more clear if presented as separate images. The green is not clear in the WT situation.

Version 1:

Reviewer comments:

Reviewer #1

(Remarks to the Author)

The authors have performed a significant amount of additional experiments to answer the referee's comments and modified

the manuscript accordingly.

I would thus recommend its publication in Nature communications in its current form.

Reviewer #2

(Remarks to the Author)

The authors have made multiple changes to the manuscript to improve it and have addressed the points I raised in the first review.

I have a few minor comments that when rectified will help the manuscript.

1. On line 151 I think it should be sup fig 1f,g not 1e,f

2. On and around line 180 the authors cite a JCB paper from Estey et al to support their statement that septins form a diffusion barrier to restrict the spread of cytoskeletal proteins in the intercellular bridge. Re-reading that paper I find that the Estey paper does not demonstrate that nor state that nor really discuss that concept. However, Estey et al in 2011 in a review in Cur. Biol. do discuss that concept based on findings in yeast and at the base of cilia. However, I am unaware of any experimental evidence to show that septins acts as a diffusion barrier in the intercellular bridge. I would recommend re-wording that section significantly and toning down that model unless there are other studies that demonstrate the concept of a diffusion barrier, which if there are they should be cited and I apologize.

3. Around lines 280-283 the authors state no effect of the treatment on the initial recruitment of anillin and septins to the furrow and cite their Sup Fig 3f. However, that figure whilst showing anillin and septins in the same cellular location +/- treatment, show cells where furrow ingression will have stopped and are about to transition into intercellular bridge assembly and elongation. I have no doubt based on the figure their statement is correct, but the figure per se does not directly support their statement. A figure with images showing a furrow would support their statement.

4. There is a problem with panel labeling in fig 5. The text refers to panels j and k which are not present but are presumably h and i

5. In sup fig 5d there is a ghost grey box that should be removed.

Reviewer #3

(Remarks to the Author)

In their rebuttal, the authors refer to a new Figure 5j. However, the revised manuscript does not have a Figure 5j, despite the legends going to all the way to k. Before spending my time on reviewing this revision, it would be great to have a proper version

Version 2:

Reviewer comments:

Reviewer #2

(Remarks to the Author)

The authors have now satisfactorily addressed all the points I have raised.

Reviewer #3

(Remarks to the Author)

The authors have properly addressed my comments

Detailed response to the referees

We would like to thank the editor(s) as well as all referees for their careful reading of our manuscript and for their thoughtful suggestions. We were very happy to learn that all three reviewers and the editor agree on the high general interest and timeliness of our study and we would like to thank them for their strong support to publish our findings in *Nature Communications*. We have now conducted a number of new experiments and made textual changes to our manuscript that fully address all questions and concerns raised in the initial round of review. We have also tackled all minor points raised as detailed in our response below (in blue).

Reviewer #1

Russo et al. present how PIPK1 γ , an enzyme synthesizing PI(4,5)P₂, interacts with partners at the intercellular bridge and midbody during cytokinesis. The results demonstrate that the isoform PIPK1 γ has a key role at the midbody as compared with other isoforms α and β . The interplay between key proteins (septin, anillin, centralspindlin) at the midbody and PIPK1 γ is described to propose a model where the synthesis of PI(4,5)P₂ tunes the organization of the midbody throughout cytokinesis. Russo et al. have done some interesting work to understand the mechanisms occurring during cytokinesis. However, some control experiments would be welcome to make their points more convincing. Hence, I would recommend some revisions before the manuscript can be accepted in *Nature Communications*.

Major revisions

- In most experiments, PIPK1 γ is depleted, while the expression, localization of possible partners are assayed. Conversely, depleting septins, anillin, or centralspindlin and describing the expression level and localization of PIPK1 γ would be key to actually understand how PIPK1 γ is controlled and dependent on those key partners. Besides, the authors state, at least twice, their assays demonstrate a “septin-dependent recruitment of PIPK1 γ ” while they actually describe a PIPK1 γ -dependent recruitment of septins.

We thank the reviewer for this valuable suggestion.

We have analyzed expression levels of PIPK1 γ upon depletion of key regulatory proteins described in our study. Our **novel Supplementary Figure 5d** demonstrates that global PIPK1 γ levels are decreased upon depletion of anillin or MgcRacGAP. This indicates that anillin- and MgcRacGAP-mediated midbody tethering to the plasma membrane is important for PIPK1 γ stability (Kechad et. al., 2012, PMID 22226749; Lekomtsev et al., 2012, PMID 23235882). We also assessed the localization of mCherry-tagged PIPK1 γ -i5 (WT) upon depletion of MgcRacGAP, or of anillin. As shown in our **novel Supplementary figure 5e**, under both knockdown conditions kinase recruitment to the ICB is significantly impaired, again in line with centralspindlin's and anillin's role in midbody tethering.

By contrast, depletion of SEPT7, a septin paralogue required for septin filament stability, does not significantly alter global expression levels of PIPK1 γ (**novel Supplementary Figure 5d**). Nonetheless, upon loss of SEPT7 mCherry-PIPK1 γ -i5 (WT) is no longer recruited to the ICB during cytokinesis (**novel Supplementary figure 5e**). This is in support of our conclusion that kinase recruitment depends on the spatiotemporally controlled deposition of septins at the ICB.

Yet, we agree that at the current stage it is difficult to unequivocally claim “septin-dependent recruitment of PIPKI γ ” and more carefully rephrased throughout the manuscript (“coincident recruitment of PIPKI γ and septins”).

- Testing the role of PIPKI on myosin II recruitment would be valuable as well given myosin is involved in cytokinesis.

As displayed in our **novel Supplementary Figure 3h** phospho-myosin-light chain recruitment to the ICB is unimpaired.

-To demonstrate synergistic roles of the different PIPKI, particularly on the distribution of PI(4,5)P₂, depletions of combinations of the isoforms would be worthwhile. Have they authors attempted to deplete combinations of 2 or all 3 simultaneously?

We do agree that PIPKI β and of PIPKI γ might play synergistic roles during furrow ingression. Based on the observation that exclusively depletion of PIPKI γ (and not depletion of PIPKI β) resulted in late mitotic defects during cytokinesis - the main focus of our study - we did not consider to deplete PIPKI β and γ in parallel. Please also see our novel data presented in **Supplementary Figure 3g**, which demonstrate that depletion of PIPKI γ -i3/i5 does not significantly impair furrow ingression, but rather inhibits timing of the microtubule cut.

- On Figure 1.h, upon depletion of PIPKI γ , septins can still be visualized bound to the plasma membrane as well as within the midbody. Was this a frequent observation? Can the authors comment on this? Besides, depletion of PIPKI β seems to also disrupt septin localization given that septins can be seen as “clusters” or point-like densities. Can the authors comment on this?

Upon depletion of PIPKI γ septins (and anillin) randomly distribute at the plasma membrane, in most cases not at the ICB itself, but rather at sites adjacent to it. Of note, the ICB itself suffers from multiple defects (reduced length: Figure 4d, reduced stability: Figure 4c, impaired MT bundling: **novel Figure 4e**). Septin clusters may also occur at plasmalemmal sites underneath the ICB, as the daughter cells are not so well separated.

Indeed, we also observe a similar tendency of septin dispersal upon depletion of PIPKI β as depicted in Figure 1i, probably due to the known function of PIPKI β in furrow ingression (Emoto et al., 2005, PMID 16162509). However, this effect was not significant.

- While using expansion microscopy septins seem to localize and organize as rings on both sides of the midbody, this localization is not seen using conventional microscopy. Instead septins seem to colocalize with microtubules in a bundle-like organization. Why would that be different?

At late stages of cytokinesis we find septins prominently enriched along the ICB, closely aligning with microtubule bundles. We suppose that the resolution limit of conventional microscopy does not allow to distinguish this strong staining from overlapping septin rings. Yet, we would like to indicate that at earlier stages, when septins start to translocate onto microtubules, we find septins also organized in ring-like structures at the midbody (see for instance Figure 3a, Supplementary Figures 3a/4g).

- From the literature, it is considered that anillin would recruit septins. However, in this study, the authors conclude that septins would promote anillin recruitment through PIPKI at the midbody. Can the authors comment on this observation?

It is generally considered that anillin recruits septins to the cleavage furrow, and remains associated with septins during cleavage furrow ingression. Based on super-resolution imaging Renshaw et al. (2014, PMID 24451548) reported that at later stages anillin associates with the midbody, and from there follows septins into ring structures flanking the midbody. They also reported that this translocation is dependent on its capability to associate with septins.

Based on our observations we conclude that depletion of PIPKI γ -i3/i5 reduces the availability of PI(4,5)P₂ at the midbody (Figure 4d/e), and prevents stable retention of anillin. Instead, anillin follows septins to aberrant PI(4,5)P₂-enriched sites in the periphery of the ICB.

- Figure 4: Depletion of PIPKI induce mis-localization of septins and disruption, to some extent, of the microtubule spindle. Is it a direct effect or does it occur through a PRC1 dependent effect? Can the authors possibly test or at least comment on any direct interaction between PRC1 and septins, for instance? Besides, did the authors consider testing how microtubule disruption would affect septin and PIPKI distribution in the cell?

We thank the reviewer for this valuable comment.

In **updated Figure 5j** we demonstrate that PRC1 co-purifies with SEPT2 in immunoprecipitation experiments. Further, in our **novel Supplementary figure S5c** we show that overexpression of eGFP-tagged PRC1 induces strong microtubule bundling in interphase cells, and that these microtubule bundles extensively recruit septins. We conclude that septins, PRC1 and centralspindlin act in concert to promote microtubule bundling along the ICB. Consistently, depletion of PIPKI γ -i3/i5 and the concomitant loss of septins from the ICB triggers defects in microtubule bundling, as quantified in our **novel Figure 4e**.

Minor revisions

- In different figures, the displayed images are too small (see, for instance, Figures 1.f, 3.d, 3.g, Supl. 1.b, Supl. 4.d, 4.e...). Enlargements would thus be welcome for the evidences to be more convincing.

We apologize for having displayed several images too small, and enlarged them.

- It would be worthwhile to show images (in supplementary) corresponding to the phenotypes described in Figure 1.d and 1.e (multipolar spindles and bridges, respectively).

Our **novel Supplementary Figure S2d** shows blowups of multipolar spindles and bridges.

- Can the authors comment on the effect of Thymidine, used for cell synchronization? Can it affect the ultrastructure, stability of the midbody and/or the recruitment of specific proteins?

Thymidine interferes with DNA synthesis, and its application arrests cells in early S-phase. Of note, thymidine was thoroughly washed out to release cells from this block. A second block was then imposed by adding nocodazole, to arrest cells in metaphase. Again, nocodazole was thoroughly washed out to let cells progress to cytokinesis. We expect that no excess of thymidine or nocodazole was present any more at the time point of fixation.

- P7, I196: The authors should specify that the affinity purification was performed from a mouse brain lysate.

We adapted the Results section accordingly.

- Figure 2e.: Some of the cells are not labelled with m-Cherry-PIP1 γ while a septin signal is visible. Can the authors give an explanation?

Cells depicted in Fig. 2e are genome-edited to express SEPT2-eGFP under its endogenous promoter. mCherry-tagged kinase variants were only transiently transfected, and the transfection efficiency was usually around 40%. Therefore, some cells are positive for SEPT2-eGFP, but negative for mCherry-tagged PIP1 γ .

- Figure 2.f: Did the authors perform the same experiment testing for the i3 isoform? Did they obtain the same phenotype as for i5 mutant?

We did not perform the same experiment with the i3 isoform. But as both isoforms co-localize with actin filaments, we expect a similar outcome. Also, in most cases we show that re-introduction of PIP1 γ -i3 is sufficient to rescue defects.

- Supplementary Fig.3: The numbering for the different panels differs from the font/numbering used for all other figures. This should be homogenized.

We apologize for this inconsistency and adapted font and numbering.

- Page 9, top (l 246-253): The described result is not displayed or it is unclear?

We apologize for this unclarity. The results are displayed in Figure 3a, as indicated in former line 246. We now added an additional reference to this figure panel in line 259.

- Figure 3.d: On the siOCRL panel, the signal from citron kinase seems much higher than in the panels above and below. However the quantification (3.f) indicates a rather constant intensity depending on the conditions. Can the authors comment or correct accordingly?

The quantification depicted in Figure 3f provides an average of 4 independent experiments, with CIT and PIP2 levels quantified from at least 15 cytokinetic cells in each experiment. Please note that for confocal image acquisition the focus was set on the midbody in the center of the ICB. In some cases this layer coincides well with the ventral surface of the cells (as in the case of the siOCRL panel depicted in Figure 3D). In most other cases, however, the midbody plane locates significantly above the dorsal surface of the cell, and the image does not appropriately reflect protein abundance in the cytoplasm.

The relative CIT-K intensity level determined from the siOCRL panel depicted in Figure 3D was 0.88, thus, a bit lower than average determined from siControl cells.

To avoid confusion we now replaced the siOCRL panel with a panel that more closely resembles average total CIT-K levels.

- Supplementary fig. 4.g: The septin pattern observed for i5 WT looks different from others and septins seem to localize in the midbody as well. Can the authors comment on this?

We thank the reviewer for his critical alertness. All depicted figure panels represent snapshots of cells fixed during cytokinesis at slightly differing stages. As reported by Renshaw septins transiently colocalize with anillin at the flanks of the midbody, before they redistribute along the

ICB. We thus assume that the septin staining depicted for i5 WT corresponds to this transition state. To avoid confusion we replaced the figure panels presented for i5 WT for a more consistent example.

- Figure 5.k: The signals for MKLP1 and septins from an IP via PIPKI γ are rather weak. Can the authors comment on this? Testing for centralspindlin, have the authors also assayed RacGap1?

The signal for MKLP1 gained from an IP via PIPKI γ are indeed weak, but we observe a significant enrichment of MKLP1 when compared to the signal from Input lanes. We found also RacGAP1 to co-immunoprecipitate with PIPKI γ in a different experiment (exp. 21-03-25; please see figure panel below).

The septin to co-purification with PIPKI γ might be impaired sterically by the antibody used for affinity purification of PIPKI γ (raised against the C-terminal tail).

Centralspindlin affinity-purifies together with PIPKI γ .

Endogenous PIPKI γ was immunoprecipitated from lysates of synchronized HeLa cells. The affinity-purified material was separated by SDS-PAGE and analyzed by Western blotting using the indicated antibodies.

Reviewer #2

The manuscript submitted by Russo and colleagues addresses the question of how PIP2 is generated at the intercellular bridge during mammalian cytokinesis and what effect it has on cytokinetic progression. They report that coupling of the PI-5-P kinase PIPKI γ with septins is required to stabilize the centralspindlin complex at the midbody/Flemming body in late telophase. This study addresses an important and unresolved question in the field and is thus of significant interest to cell division researchers. Additionally, it reveals an interesting coupling of the septin cytoskeleton to lipid synthesis. However, a number of clarifications are required.

We thank the reviewer for appreciating the significance of our study.

Major points:

1. A major deficiency in the manuscript is a lack of rescues for siRNA-mediated depletions in multinucleate experiments described in Figures 1 and Supp Figure 3D. Given that the phenotypes are small in absolute terms, these rescues are particularly important to establish that the relatively small increases observed for cytokinesis failure are not due to off-target effects. While the rescues shown in Fig 3i are appreciated, they are not direct evidence of a

rescue of cytokinesis failure. Rescues would preferably be undertaken with strategies that avoid over-expression.

This is a valid concern.

We performed rescue experiments using stably transfected cell lines expressing mCherry, or siRNA-resistant, mCherry-tagged PIPKI γ -i5 (WT). As depicted in our **novel Supplementary Figure 3e** depletion of endogenous PIPKI γ -i3/i5 significantly increased the fraction of multinucleated cells in mCherry-expressing cells, but not in cells expressing mCherry-tagged PIPKI γ -i5 (WT). We these data unequivocally demonstrate that cytokinesis failure is not an off-target effect of the siRNA used throughout our experiments.

2. The phenotype induced by PIPKI γ disruption (either by siRNA alone or with expression of mutants) requires more defining. If PIPKI γ is important for telophase progression, is there a measurable delay in telophase? Does abscission take longer? Some timing experiments could help clarify

We agree that timing experiments are helpful to dissect between defects occurring in early or late cytokinesis.

We used live cell imaging of cells labelled with SiR-tubulin, to follow microtubule dynamics in control and PIPKI γ -i3/i5-depleted cells. This approach allowed us to measure the time from initiation of mitosis (marked by cell rounding) until completion of furrow ingression (marked by the appearance of a thin microtubule bridge), and thereafter until microtubule cut (marked by a sudden retraction of the microtubule bundle from the ICB into one of the daughter cells). Our analyses (depicted in **novel Supplementary Figure 3g**) revealed no significant change in furrow ingression in kinase-depleted cells, but indicated a significant delay of the microtubule cut.

3. Some terms used in the manuscript are unclear and ambiguous. The common one of 'compact' to describe anillin localization is unclear and does not reflect what can be seen in the images. (Fig 1,3). Nor does this reflect previous studies that defined anillin dynamics in the intercellular bridge (El Amine et al. 2013, Renshaw et al. 2014). These two studies characterized in detail anillin dynamics and define common organizational states that reflect different phases of intercellular bridge assembly. Assessing if and/or when these previously defined patterns of anillin localization are disrupted by the various perturbations performed in this study will make comparisons across studies much easier.

We thank the reviewer for pointing this out. We now phrase less ambiguously to describe the localization of anillin ("cells with anillin at the ICB"). Furthermore, we assessed the localization of anillin also at earlier stages of cytokinesis. In line with our observation that PIPKI γ -i3/i5 have no obvious effect on furrow ingression time we find septins and anillin associated with the ingressing furrow (**novel Supplementary figure 3f**) in control and in kinase-depleted cells. Likewise, active myosin as evaluated by immunostainings of phospho-myosin LC remains largely unaffected by depletion of PIPKI γ -i3/i5 (**novel Supplementary Figure 3h**).

4. Related to point 3, in many cases the micrographs being compared appear to be at different stages of intercellular bridge development (i.e. that the authors are comparing two ICBs of differing ages). This occurs frequently in Figs 3 , S3 and 4. Including tubulin staining to the experiments would help clarify these issues as the degree of microtubule compaction has been used to assess the intercellular bridge age thereby allowing more direct "apples to apples" comparisons. This is a very important point with respect to the conclusions made and should be rigorously addressed.

We agree that under control conditions the appearance of the acetylated tubulin staining is indicative of the maturation state of the ICB. However, our live experiments (Movies 1 and 2), in which we monitored the distribution of eGFP-SEPT2 in genome-edited cells, clearly show that the ICB between the two daughter cells never elongates, irrespective of how long we wait. We, therefore, feel that the appearance of acetylated tubulin bridge is not an adequate marker for the maturation state of the bridge, but rather a defect caused by the abolished recruitment of septins onto ICB microtubules, and the concomitant destabilization of the microtubule bundle.

To more closely illustrate this phenotype we measured the width of the microtubule bundle at a distance of 4 μm apart from the center of the ICB. Our **novel Figure 4e** clearly shows that depletion of PIPKI γ -i3/i5 impairs bundling in the periphery of the ICB, and that this defect is rescued by introduction of siRNA-resistant PIPKI γ -i3 or i5 (WT), but not of kinase-dead or of septin binding-deficient variants.

Minor points:

A. In the introduction the Piekny and Maddox review is extensively cited. While this is a good review it is 15 years old and much has been learnt subsequently about anillin, who it binds to and how it functions. It would be useful to newcomers to the field if more up to date references about specific points were used.

We followed the reviewer's suggestion and updated the references for anillin functions.

B. The biology of septins needs a more thorough treatment, again with up to date references. While it is said that septins hetero-oligomerize, the hexamer/octamer distinction is bound to be confusing to non-experts and should be better developed. In addition, in the results section (line 170) the authors state that septins bind to the C-term of anillin. While this is the conclusion drawn from this 2000 paper, subsequent work suggests a more complex relationship. This combined with a better discussion of hexamers and octamers would bring the paper up to date and allow a subsequent more detailed discussion (and perhaps experimental plan) of the underlying mechanism at play.

We thank the reviewer for pointing this out and also updated information about the interplay between septin and anillin.

C. In the results, a simple statement that cells were synchronized by double thy + nocodazole treatment should be added to the main text to aid the reader.

We included a brief statement in the Results section to aid the reader.

D. Experiments demonstrating the localization of anillin and septins in PIPKI γ -depleted cells across the stages of cytokinesis (as shown for control cells in Fig 3a) would be helpful. As the authors state, anillin and septin recruitment to the equatorial membrane prior to furrow ingression are known to be dependent on PI(4,5)P2. However, the manuscript only shows disrupted telophase localization of anillin in knockdown/mutant conditions. This may result from deficiencies in recruitment to the furrow/ICB in anaphase-early telophase as opposed to the action of PIPKI γ later in telophase.

This relates to the concern raised in point 3. As indicated above we find septins and anillin associated with the ingressing furrow (**novel Supplementary figure 3f**) in control and in kinase-depleted cells. Likewise, active myosin as evaluated by immunostainings of phosphor-myosin LC remains largely unaffected by depletion of PIPKI γ -i3/i5 (**novel Supplementary Figure 3h**). We conclude that the interplay between septins and PIPKI γ -i3/i5 does not affect anaphase-to-telophase transition, but rather affects ICB morphology and stability during late cytokinesis. But we note, that we cannot rule out additional effects of other isoforms of PIPKI γ . These would need to be addressed in future studies.

E. What effect does knockdown of PIPKI α & β have on PI(4,5)P₂ levels in anaphase and telophase cells? The results of Fig 3 would suggest that PIPKI γ is responsible for the majority of PI(4,5)P₂ present in cytokinetic cells. Is the remaining PI(4,5)P₂ sufficient to promote initial anillin/septin ICB recruitment?

Based on the observation that loss of PIPKI γ -i3/i5 does not impact on anillin (and concomitant septin) recruitment to the early ICB we conclude that other type I PIPK enzymes (PIPKI α and/or PIPKI β and/or other isoforms of PIPKI γ) provide a sufficiently large pool of PI(4,5)P₂ to allow for unimpaired furrow ingression.

F. Fig 4: Measurements comparing the width of ICB bridges across conditions should be performed. If as the authors argue, depletion of PIPKI γ -i3-5 perturbs the development of the ICB such that its microtubules do not compact as much as control, then the width of acetylated tubulin staining will better bolster this point than measurements of ICB length.

This relates to major concern 4.

We measured the width of the microtubule bundle in the center of ICB, and at a distance of 4 μ m apart. Our **novel Figure 4e** clearly shows that depletion of PIPKI γ -i3/i5 impairs bundling in the periphery of the ICB, and that this defect is rescued by introduction of siRNA-resistant PIPKI γ -i3 or i5 (WT), but not of kinase-dead or of septin binding-deficient variants. Please note that the acetylated tubulin staining needed to be boosted to visualize “fluffy” MTs for quantifications, which are not so apparent in the figure panels presented throughout the manuscript.

G. Fig5hi: I do not think it's appropriate to measure fluorescence intensities across different expanded samples unless steps were taken to ensure some normalization of expansion factors and a comparison to some constant. The obvious differences in intensity presented in panel g are more than sufficient to support the authors' claims.

As indicated in the Methods section we did calculate expansion factors to ensure comparability between different conditions and specimen. Yet, we agree that the differences in intensity depicted are obvious, and we removed the quantification.

H. In S3G: the i5 Δ supSB panel contains a cell that appears to be at an earlier stage of cytokinesis compared to the other cells in this panel (based on less compaction of tubulin staining). It further also looks to be multinucleated. Is this the norm? If yes this would have implications in interpretation.

We assume that the reviewer refers to Supplementary Figure 4g. As delineated in our response to point 4 we would like to stress again that the defective ICB is part of the phenotype. None

of the stably transfected cell lines expressing mCherry or mCherry-tagged kinase variants displays elevated levels multinucleation (see also quantification in **novel Supplementary Figure 3e**). Hence, to better represent our data we exchanged multinucleated cells in Supplementary Figure 4g (i5 Δ SB) and in Supplementary Figure 3j (i5 K188A) and (i5 Δ SB).

I. line 76 PI(4,5)2 should be PI(4,5)P2

We thank the reviewer for carefully reading the manuscript. We corrected accordingly.

Reviewer #3

This manuscript studies the spatiotemporal control of PI(4,5)P2 during cytokinesis and reports a key role for two splice isoforms of PIPKI γ . The authors demonstrate that PIPKI γ is required for cytokinesis progression and for the organization of anillin and septins at the intracellular bridge (Figure 1). They furthermore identify two splice isoforms (i3 and i5) of PIPKI γ that can interact with septins (Figure 2) and demonstrate that these are critical for proper anillin accumulation and PI(4,5)P2 synthesis near the midbody (Figure 3), as well as for the recruitment of septin to the intracellular bridge (Figure 4) and the association of the central spindle complex to the midbody (Figure 5). Based on these results, the authors propose a model in which septins recruit PIPK γ to the cleavage furrow and generates local PI(4,5)P2, whereas later on septin-bound PIPK γ create even more local PI(4,5)P2 that contributes to septin retention and subsequent microtubule stabilization. Overall, the data is convincing and the writing is clear. Nonetheless, I do have a number of comments that need to be properly addressed to make the manuscript ready for publication.

We thank the reviewer for appreciating the validity of our data.

1. I found the authors conclusions on the interdependence of septins and PIPKI γ unclear. Initially, the authors postulate that septin recruits PIPKI γ to the midzone, whereas later on they suggest that PIPKI γ is key for proper localization of SEPT6, including its localization to microtubules. The text and cartoons should be improved to clarify how the authors think about this interdependence.

As indicated in our response to comment 1 of reviewer 1 we assessed the distribution of mCherry-tagged PIPKI γ -i5 (WT) under different knockdown conditions, and indeed find that kinase recruitment to the late stage ICB is impaired upon depletion of SEPT7 (a septin paralog essential for the assembly of septins into filaments (**novel Supplementary Figure 5d**). Likewise, mCherry-PIPKI γ -i5 (Δ SB) is not properly recruited to the ICB (Supplementary Figures 3j and 4g). Based on these findings we conclude that septins are required for localizing septin binding PIPKI γ isoforms to the ICB. We modified cartoons and text and hope this interdependence is now clearer.

As anillin/septin recruitment to the cleavage furrow and furrow ingression time (**novel Supplementary Figures 3f/g**) remained unaffected by kinase depletion, we did not analyze in depth the early-stage localization of PIPKI γ at the cleavage furrow. Yet, we do see coincident recruitment of septins and mCherry-PIPKI γ -i5 (WT) to the anillin-enriched cleavage furrow, and re-phrased accordingly.

2. In addition, the whole interplay between septins, PIPKI γ and microtubule stabilization is not well developed and therefore not very convincing. How does production of PI(4,5)P2 helps to enrich septins on microtubules?

Novel data depicted in **updated Figure 5j** and **novel Supplementary Figure 5e** demonstrate that septins form an endogenous complex with PRC1 and localize to eGFP-PRC1-induced microtubule bundles in transfected fibroblasts. Our data indicate that septins at the ingressing cleavage furrow form complex with centralspindlin as soon as the furrow approaches the microtubule bundle at the center of the ICB. Septin-bound PIPKI γ -isoforms also associate with centralspindlin, and this initiates PI(4,5)P₂ synthesis at the midbody (maybe fostered by centralspindlin-bound Arf6, a known activator of PIPKI γ , Krauss et al., 2003, PMID 12847086) to stably tether centralspindlin to the plasma membrane. Upon completion of furrow ingression septins are released from the plasma membrane by an unknown mechanism, and instead associate with centralspindlin-/PRC1-positive microtubules.

In absence of PIPKI γ -i3/i5 PI(4,5)P₂ levels at the midbody are too low to stably anchor centralspindlin, leading to impaired microtubule bundling along the ICB, and a loss of PRC1. Therefore, upon release from the plasma membrane septins are not retained on microtubules but rather accumulate at PI(4,5)P₂-enriched sites in the periphery of the ICB.

We have adapted the model depicted in **Figure 6** according to our novel findings, and also updated the Discussion.

3. The title and discussion put a lot of emphasis on nanoscale synthesis, but it is unclear if this means anything else than localized, restricted synthesis. Of course, if an individual protein modifies lipids this can always be considered nanoscale synthesis, but the current formulations are raising the expectations that the authors have discovered nanosized membrane regions that carry specific modification. This is not at all reflected in the presented data.

We agree, and rephrased.

Minor comments

4. Figure S3 – panel labels are inconsistent with all other figures

We thank the reviewer for his awareness. We corrected the panel labels.

5. Figure 2f – The merge images of SEPT2 and PIPK γ are confusing and would be more clear if presented as separate images. The green is not clear in the WT situation.

We included separate images in the figure panels to more clearly indicate the localization of SEPT2 and PIPKI γ .

Detailed response to the referees

We would like to thank the editor as well as all referees once again for their careful reading of our manuscript and for their thoughtful suggestions. We were very happy to learn that all three reviewers and the editor agree on the high general interest and timeliness of our study and we would like to thank them for their strong support to publish our findings in *Nature Communications*. We have now made textual changes to our revised manuscript that fully address all questions and concerns raised in the second round of review as detailed in our response below (in blue).

Reviewer #1:

The authors have performed a significant amount of additional experiments to answer the referee's comments and modified the manuscript accordingly. I would thus recommend its publication in Nature communications in its current form.

We thank the reviewer for his positive decision!

Reviewer #2

The authors have made multiple changes to the manuscript to improve it and have addressed the points I raised in the first review.

We thank the reviewer for appreciating the improvements of our manuscript.

I have a few minor comments that when rectified will help the manuscript.

1. On line 151 I think it should be sup fig 1f,g not 1e,f.

We apologize for this mistake, and corrected accordingly.

2. On and around line 180 the authors cite a JCB paper from Estey et al to support their statement that septins form a diffusion barrier to restrict the spread of cytokinetic proteins in the intercellular bridge. Re-reading that paper I find that the Estey paper does not demonstrate that nor state that nor really discuss that concept. However, Estey et al in 2011 in a review in *Cur. Biol.* do discuss that concept based on findings in yeast and at the base of cilia. However, I am unaware of any experimental evidence to show that septins acts as a diffusion barrier in the intercellular bridge. I would recommend re-wording that section significantly and toning down that model unless there are other studies that demonstrate the concept of a diffusion barrier, which if there are they should be cited and I apologize.

In their study Estey et al. demonstrate that SEPT9 is required for exocyst deposition at the midbody, where the two are found adjacent to each other, rather than being colocalized. We agree that the authors do not provide direct evidence that septins act as a diffusion barrier at the ICB. We, thus, re-phrased as follows (lines 188-191):

Although largely dispensable during furrow ingression in mammalian cells, septins exert pivotal functions once the ICB is established and extends by compartmentalizing select cytokinetic proteins at the midbody and by scaffolding the assembly of the abscission machinery.

3. Around lines 280-283 the authors state no effect of the treatment on the initial recruitment of anillin and septins to the furrow and cite their Sup Fig 3f. However, that figure whilst showing anillin and septins in the same cellular location +/- treatment, show cells where furrow ingression will have stopped and are about to transition into intercellular bridge assembly and elongation. I have no doubt based on the figure their statement is correct, but the figure per se does not directly support their statement. A figure with images showing a furrow would support their statement.

We thank the reviewer for pointing this out. We added an additional panel depicting anillin and septin colocalization at the cleavage furrow also at early telophase, and demonstrate that this is independent of presence or absence of PIPKI γ -i3/i5 (please see modified **Supplementary Figure 3f**).

4. There is a problem with panel labeling in fig 5. The text refers to panels j and k which are not present but are presumably h and i.

We apologize for the mistake, and corrected figure numbering and the legend accordingly.

5. In sup fig 5d there is a ghost grey box that should be removed.

We did not find a ghost grey box in Supplementary Figure 5d, but in Figure 2f and in Supplementary 2d, which we removed.

Reviewer #3

In their rebuttal, the authors refer to a new Figure 5j. However, the revised manuscript does not have a Figure 5j, despite the legends going to all the way to k. Before spending my time on reviewing this revision, it would be great to have a proper version.

We deeply apologize for the inaccurate figure numbering, which is now corrected in the revised version of our manuscript.

We include once again our response to this reviewer's initial comments:

This manuscript studies the spatiotemporal control of PI(4,5)P₂ during cytokinesis and reports a key role for two splice isoforms of PIPKI γ . The authors demonstrate that PIPKI γ is required for cytokinesis progression and for the organization of anillin and septins at the intracellular bridge (Figure 1). They furthermore identify two splice isoforms (i3 and i5) of PIPKI γ that can interact with septins (Figure 2) and demonstrate that these are critical for proper anillin accumulation and PI(4,5)P₂ synthesis near the midbody (Figure 3), as well as for the recruitment of septin to the intracellular bridge (Figure 4) and the association of the central spindle complex to the midbody (Figure 5). Based on these results, the authors propose a model in which septins recruit PIPKI γ to the cleavage furrow and generates local PI(4,5)P₂, whereas later on septin-bound PIPKI γ create even more local PI(4,5)P₂ that contributes to septin retention and subsequent microtubule stabilization. Overall, the data is convincing and the writing is clear. Nonetheless, I do have a number of comments that need to be properly addressed to make the manuscript ready for publication.

We thank the reviewer for appreciating the validity of our data.

1. I found the authors conclusions on the interdependence of septins and PIPKlg unclear. Initially, the authors postulate that septin recruits PIPKlg to the midzone, whereas later on they suggest that PIPKlg is key for proper localization of SEPT6, including its localization to microtubules. The text and cartoons should be improved to clarify how the authors think about this interdependence.

As indicated in our response to comment 1 of reviewer 1 we assessed the distribution of mCherry-tagged PIPK γ -i5 (WT) under different knockdown conditions, and indeed find that kinase recruitment to the late stage ICB is impaired upon depletion of SEPT7 (a septin paralog essential for the assembly of septins into filaments (**novel Supplementary Figure 5e**). Likewise, mCherry-PIPK γ -i5 (Δ SB) is not properly recruited to the ICB (Supplementary Figures 3k and 4g). Based on these findings we conclude that septins are required for localizing septin binding PIPK γ isoforms to the ICB. Later, PIPK γ -i3/i5-dependent synthesis of PI(4,5)P₂ is required to facilitate centralspindlin- and PRC1-dependent translocation of septins onto microtubules. Specifically, centralspindlin, by tethering bridge microtubules to the furrow membrane, may bring septins into close proximity with the cytokinetic bridge and thereby facilitate their translocation. PRC1, on the other hand, may assist septin translocation onto bridge microtubules by directly interacting with septins (please also see our response to point 2 for further clarification). We modified cartoons and text and hope our view of this interdependence becomes now clearer.

As anillin/septin recruitment to the cleavage furrow and furrow ingression time (**novel Supplementary Figures 3f/g**) remained unaffected by kinase depletion, we did not analyze in depth the early-stage localization of PIPK γ at the cleavage furrow. Yet, we do see coincident recruitment of septins and mCherry-PIPK γ -i5 (WT) to the anillin-enriched cleavage furrow, and re-phrased accordingly.

2. In addition, the whole interplay between septins, PIPKlg and microtubule stabilization is not well developed and therefore not very convincing. How does production of PI(4,5)P₂ helps to enrich septins on microtubules?

Novel data depicted in **updated Figure 5h** and **novel Supplementary Figure 5c** demonstrate that septins form an endogenous complex with PRC1 and localize to eGFP-PRC1-induced microtubule bundles in transfected HeLa cells. Our data indicate that septins at the ingressing cleavage furrow form complex with centralspindlin as soon as the furrow approaches the microtubule bundle at the center of the ICB. Septin-bound PIPK γ -isoforms also associate with centralspindlin, and this initiates PI(4,5)P₂ synthesis at the midbody (maybe fostered by centralspindlin-bound Arf6, a known activator of PIPK γ , Krauss et al., 2003, PMID 12847086) to stably tether centralspindlin to the plasma membrane. Upon completion of furrow ingression septins are released from the plasma membrane by an unknown mechanism, and associate with centralspindlin-/PRC1-positive microtubules. Presence of septins and PRC1 may further favor their bundling.

In absence of PIPK γ -i3/i5 the retention of centralspindlin at the midbody is impaired, as is the translocation of septins onto microtubules. Of note, PRC1 levels decline concomitantly, which may reflect a mutual stabilization of septins and PRC1 on bundled microtubules. As a consequence, upon release from the plasma membrane septins are not retained on microtubules, but rather accumulate at PI(4,5)P₂-enriched sites in the periphery of the ICB. We have adapted the model depicted in **Figure 6** according to our novel findings, and also updated the Discussion.

3. The title and discussion put a lot of emphasis on nanoscale synthesis, but it is unclear if this means anything else than localized, restricted synthesis. Of course, if an individual protein

modifies lipids this can always be considered nanoscale synthesis, but the current formulations are raising the expectations that the authors have discovered nanosized membrane regions that carry specific modification. This is not at all reflected in the presented data.

We agree, and rephrased.

Minor comments

4. Figure S3 – panel labels are inconsistent with all other figures

We thank the reviewer for his awareness. We corrected the panel labels.

5. Figure 2f – The merge images of SEPT2 and PIPK γ are confusing and would be more clear if presented as separate images. The green is not clear in the WT situation.

We included separate images in the figure panels as Supplementary Figure 2d to more clearly indicate the localization of SEPT2 and PIPK γ .